# Abnormal calcium release and delayed afterdepolarizations: A comparison of two mathematical models for human ventricular myocytes

**Navneet Roshan**[ID]*, **Rahul Pandit**[ID]

Centre for Condensed Matter Theory, Department of Physics, Indian Institute of Science, Bangalore, India

* navneet@iisc.ac.in

## Abstract

Focal arrhythmias, which arise from delayed afterdepolarizations (DADs), are observed in various pathophysiological heart conditions; these can lead to sudden cardiac death. A clear understanding of the interplay of electrophysiological factors of cardiac myocytes, which lead to DADs, can suggest pharmacological targets that can eliminate DAD-induced arrhythmias. Therefore, we carry out multiscale investigations of two mathematical models for human-ventricular myocytes, namely, the ten Tusscher-Panfilov TP06 model and the HuVEC15 model of Himeno, *et al.*, at the levels of single myocytes, one- and two-dimensional (1D and 2D) tissue, and anatomically realistic bi-ventricular domains. We demonstrate that the Sarco/endoplasmic reticulum $Ca^{2+}$-ATPase (SERCA) pump uptake rate and the $Ca^{2+}$ leak through the ryanodine-receptor (RyR) channel impact this transition significantly. We show that the frequencies and amplitudes of the DADs are key features that can be used to classify them into three types, at the single-myocyte level. By carrying out detailed parameter-sensitivity analyses, we identify the electrophysiological parameters, in the myocyte models, that most affect these key features. We then obtain stability (or phase) diagrams that show the regions of parameter space in which different types of DADs occur. By comparing differences in model compartmentalizations, we show that these structural features can significantly influence both the occurrence and the types of DADs. We demonstrate in the TP06 model, the $Na^+/Ca^{2+}$ exchanger can also play a protective role in the elimination of DADs, and the presence of late calcium releases can enhance this effect. We present representative tissue simulations of the spatiotemporal evolution of waves of electrical activation, in these models, to illustrate how arrhythmogenic premature ventricular complexes (PVCs) emerge from patches of DAD cells, when we pace the tissue.

**Data availability statement:** Find the code here: https://doi.org/10.5281/zenodo.14360369 All relevant data are within the manuscript and its Supporting Information files.

**Funding:** The author(s) received no specific funding for this work.

**Competing interests:** NO authors have competing interests.

# 1 Introduction

Cardiac diseases are the leading cause of mortality in industrialized nations [1], with afterdepolarizations driven arrhythmias one of the contributers to this burden. Afterdepolarizations are abnormal increases in the transmembrane potential ($V_m$) of cardiac myocytes, and are categorized as early (EADs) or delayed (DADs) depending on whether they occur during the repolarization or diastolic phase of the action potential, respectively [2–4]. In this study, we focus on DADs, which are commonly observed across various cell types [5–12], and are implicated in a range of clinical conditions such as catecholaminergic polymorphic ventricular tachycardia (CPVT) [13], nonischemic heart failure [14], and digitalis toxicity [15,16].

The primary cellular mechanism underlying DADs is spontaneous calcium release (SCR) from the sarcoplasmic reticulum (SR), typically triggered by intracellular calcium overload [17]. In experiments, this overload is induced by various means [18, 19], while in mathematical models, it can be simulated by increasing the conductance of the L-type calcium current ($I_{CaL}$) [20]. The SCRs elevate cytosolic calcium, activating the Na$^+$/Ca$^{2+}$ exchanger (NCX), which generates a depolarizing current; and the SERCA pump, which returns calcium to the SR. This interplay modulates $V_m$ and shapes the amplitude and threshold behavior of DADs [21].

To investigate how subcellular factors influence delayed afterdepolarizations (DADs) and their potential to trigger premature ventricular complexes (PVCs), we performed a comparative analysis of two biophysically detailed human ventricular myocyte models: the ten Tusscher–Panfilov 2006 (TP06) model [22], which has been widely used in cardiac electrophysiology studies, and the more recent HuVEC15 model [23], developed with a greater emphasis on intracellular calcium compartmentalisations. Both models satisfy the calcium-oscillation hypothesis [20], which proposes that DADs arise from spontaneous calcium releases (SCRs) from the sarcoplasmic reticulum (SR) under conditions of calcium overload. Importantly, both models incorporate a depletion-dependent mechanism for SR calcium release termination [24], but do not include an explicit inactivation mechanism for ryanodine receptors (RyRs)—a simplification commonly adopted in common-pool models. Under appropriate conditions, both TP06 and HuVEC15 are capable of exhibiting DAD-like behavior, making them suitable platforms for exploring the subcellular determinants of DADs.

The HuVEC15 model provides a more refined representation of L-type calcium channels (LCCs) and RyRs, enabling a more accurate description of localized calcium-induced calcium release (CICR). It also incorporates detailed calcium compartmentalization, dividing the cytosolic space into multiple subdomains such as junctional space (jnc), intermediate zone (iz), and bulk cytosol (blk), and further subdividing the SR. Ion channels such as the Na$^+$/Ca$^{2+}$ exchanger (NCX) and LCCs are distributed across these compartments, allowing for spatially heterogeneous electrogenic effects and intracellular calcium handling [25].

Within this framework, we systematically examine how physiological factors including calcium load, SR leak, NCX activity, and compartmental diffusion influence SCRs and the resulting membrane depolarizations. By comparing TP06 and HuVEC15 under equivalent stimulation and Ca overload protocols, we aim to identify the key determinants of DAD initiation, amplitude, and their ability to propagate and trigger PVCs in tissue-level simulations. This comparative approach highlights how model structure and complexity shape the emergence of arrhythmogenic events and helps define the minimal requirements necessary to reproduce diverse forms of DAD behavior.

We perform multiscale *in silico* simulations, from single-cell to tissue levels, in both idealized and anatomically realistic domains, to explore the parameter regimes that generate DADs and PVCs in each model, identify the key features that differentiate the model response to calcium overload, and classify DADs into distinct types based on their amplitude, timing, and triggering potential.

We have organized the rest of this paper as follows. In Sect 2 we describe the models we use and the numerical and theoretical methods that we employ. Sect 3 is devoted to a detailed presentation of our results. Sect 4 discusses our results in the context of earlier numerical and experimental studies.

## 2 Models and methods

### 2.1 Electrophysiology models

To describe the human-ventricular-myocyte action potential (AP) and its $Ca^{2+}$ subsystem, we use the TP06 [22] and HuVEC15 [23] mathematical models, which adopt different approaches for the modeling of the $Ca^{2+}$ subsystem. The schematic diagrams in Fig 1 illustrate the differences between these models.

The TP06 and HuVEC15 models account for 12 and 14 transmembrane ionic currents, respectively. We use the following ordinary differential equation (ODE) for the single-myocyte $V_m$:

$$\frac{dV_m}{dt} = -\frac{I_{stim} + I_{model}}{C_m};$$

(1)

and we use the following partial differential equation (PDE) for the spatio-temporal evolution of $V_m$ in a monodomain model for cardiac tissue:

$$\frac{\partial V_m}{\partial t} = -\frac{I_{stim} + I_{model}}{C_m} + D\nabla^2 V_m;$$

(2)

here, $t$ is the time, $C_m$ is the capacitance per unit area of the myocyte membrane, $I_{stim}$ is the externally applied current stimulus to the myocyte, $I_{model}$ is the sum of all transmembrane ionic currents in the model, and $D$ is the diffusion constant, which is taken to be a scalar (isotropic) for simplicity, except when we employ an anatomically realistic bi-ventricular domain. For the TP06 and HuVEC15 models, we use, respectively,

$$\begin{aligned} I_{TP06} &= I_{Na} + I_{CaL} + I_{K1} + I_{Kr} + I_{Ks} + I_{to} \\ &+ I_{pK} + I_{bCa} + I_{NaCa} + I_{NaK} + I_{bNa} + I_{pCa} \end{aligned}$$

(3)

and

$$\begin{aligned} I_{HuVEC15} &= I_{Na} + I_{CaL} + I_{K1} + I_{Kr} + I_{Ks} + I_{Kto} + I_{Kpl} \\ &+ I_{Cab} + I_{NCX} + I_{NaK} + I_{KATP} + I_{PMCA} \\ &+ I_{l(Ca)} + I_{bNSC}; \end{aligned}$$

(4)

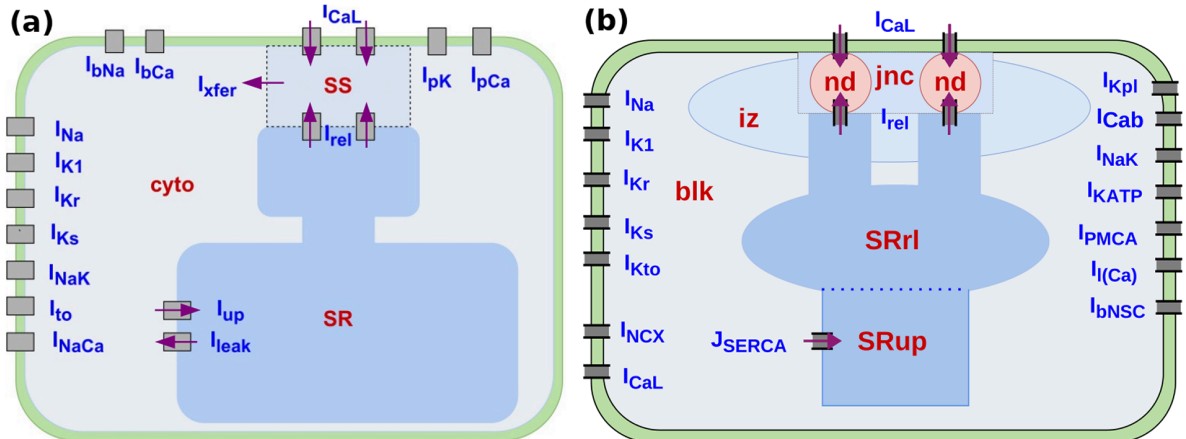

**Fig 1. Schematic diagrams comparing the Ca²⁺ subsystems and compartmentalisations in the TP06 and HuVEC15 human-ventricular-myocyte models.** (a) The TP06 myocyte volume is divided into three Ca²⁺ compartments: the sub-space (SS), the sarcoplasmic reticulum (SR), and the cytosol (CYTO); $I_{rel}$ is the Ca²⁺ release rate from the SR to the SS; the RyRs and $I_{CaL}$ open into the SS. $I_{up}$ is the SERCA pump uptake rate of Ca²⁺ from the CYTO to the SR; $I_{leak}$ is the Ca²⁺ leak from the SR to the CYTO; and $I_{xfer}$ is the diffusion flux of Ca²⁺ ions from the SS to the CYTO; $I_{NCX}$ and $I_{CaL}$ are transmembrane currents. (b) The HuVEC15 is a tightly coupled $I_{CaL}$ and RyR model and it has more Ca²⁺ compartments than there are in the TP06 model; these are the junctional space (jnc), the intermediate zone (iz), the nano domain (nd), and the bulk space (blk). The sarcoplasmic reticulum (SR) has two sub-compartments, namely, $SR_{up}$ and $SR_{rl}$, with Ca²⁺ concentrations $[Ca^{2+}]_{SR_{up}}$ and $[Ca^{2+}]_{SR_{rl}}$, respectively; the jnc is the region where the RyR and $I_{CaL}$ channel openings meet; $J_{SERCA}$ is the Ca²⁺ uptake rate from the CYTO to the SR; $I_{NCX}$ and $I_{CaL}$ are transmembrane currents, that are part of Ca²⁺- subsystem.

the currents for the TP06 (Eq 3) and HuVEC15 (Eq 4) models are defined in Table 1; and Refs. [22,23] describe in detail the ODEs for ion-channels and gating variables in the TP06 and HuVEC15 models, respectively.

Except for a few minor differences, both models share most ion channels; however, their Ca²⁺ handling differs markedly. In the TP06 model, the sarcoplasmic reticulum (SR) is represented as a single compartment, and the intracellular space is divided into the subspace (SS) and the cytosol (CYTO). By contrast, the HuVEC15 model partitions the SR into two regions: the uptake compartment, $SR_{up}$, and the release compartment, $SR_{rl}$. These are coupled by diffusion flux from $SR_{up}$ to $SR_{rl}$. Moreover, HuVEC15 features more detailed intracellular compartmentalization: the cytosol is subdivided into junctional (jnc), intermediate zone (iz), and bulk cytosol (blk) compartments. This level of detail permits modeling of heterogeneous ion-channel distribution across compartments; for example, NCX and LCCs are nonuniformly distributed in HuVEC15.

For calcium release via ryanodine receptors (RyRs), TP06 employs a reduced version of the calcium-induced calcium release (CICR) model described in Refs. [26,27]. In contrast, HuVEC15 uses a model based on the tight coupling between the LCC current ($I_{CaL}$) and RyRs [28]. In both models, RyR opening depends on the calcium concentration in the subspace ($[Ca^{2+}]_{SS}$) or nano-domain, as well as on calcium in the SR store (i.e., $[Ca^{2+}]_{SR}$ in TP06 and $[Ca^{2+}]_{SRrl}$ in HuVEC15). Notably, neither TP06 nor HuVEC15 includes explicit RyR inactivation; instead, termination of SR calcium release depends on depletion of calcium in the respective SR release store [24].

## 2.2 Numerical integration of ODEs and PDEs

The formulation of ion channel dynamics differs between the TP06 and HuVEC15 models. The gating variables described using the Hodgkin–Huxley formalism were integrated with the Rush–Larsen method, whereas the ODEs of other forms are integrated using the generalized Rush–Larsen scheme (see Refs. [29,30] for comparison).

**Table 1. Lists of the ionic currents used in the TP06 and the HuVEC15 models for human-ventricular myocytes; Refs. [22] and [23] describe the details of the ODEs for the TP06 and HuVEC15 models, respectively.**

| TP06 currents | | HuVEC15 currents | |
|---|---|---|---|
| $I_{Na}$ | Fast Na$^+$ | $I_{Na}$ | Na$^+$ (Fast and Late) |
| $I_{CaL}$ | L-type Ca$^{2+}$ | $I_{CaL}$ | L-type Ca$^{2+}$ |
| $I_{K1}$ | Inward rectifier K$^+$ | $I_{K1}$ | Inward rectifier K$^+$ |
| $I_{Kr}$ | Rapid delayed rectifier K$^+$ | $I_{Kr}$ | Fast delayed rectifier K$^+$ |
| $I_{Ks}$ | Slow delayed rectifier K$^+$ | $I_{Ks}$ | Slow delayed rectifier K$^+$ |
| $I_{to}$ | Transient outward K$^+$ | $I_{Kto}$ | Transient outward K$^+$ |
| $I_{pK}$ | Plateau K$^+$ | $I_{Kpl}$ | Plateau K$^+$ |
| $I_{bCa}$ | Background Ca$^{2+}$ | $I_{Cab}$ | Background Ca$^{2+}$ |
| $I_{NaCa}$ | Na$^+$-Ca$^{2+}$ exchanger | $I_{NCX}$ | Na$^+$-Ca$^{2+}$ exchanger |
| $I_{NaK}$ | Na$^+$- K$^+$ ATPase | $I_{NaK}$ | Na$^+$-K$^+$ pump |
| $I_{bNa}$ | Na$^+$ background | $I_{KATP}$ | ATP-sensitive potassium |
| $I_{pCa}$ | Plateau Ca$^{2+}$ | $I_{PMCA}$ | Plasma membrane Ca$^{2+}$ -ATPase |
| | | $I_{I(Ca)}$ | Ca$^{2+}$-activated background cation |
| | | $I_{bNSC}$ | Background non-selective cation |

The time-step we use to integrate the above ODEs and PDEs is 0.02 ms; and the spatial-grid size is 0.025 cm. Here, the diffusion constant is $D = 0.00154$ cm$^2$/s, for the TP06 model, and $D = 0.0012$ cm$^2$/s, for the HuVEC15 model; these values lead to conduction velocities of 67 cm/s and 62 cm/s for the TP06 and HuVEC15 models, respectively. We employ three-, five-, and seven-point stencils for the Laplacians in our one-dimensional (1D), two-dimensional (2D), and three-dimensional (3D) simulations. In our 1D, 2D, and 3D simulations, we use the following domain sizes, respectively: a cable with 256 grid points; a rectangle with $512 \times 220$ grid points that is $12.8 \times 5.5$ cm$^2$; and a cuboid with $512 \times 220 \times 20$ grid points that is $12.8 \times 5.5 \times 2$ cm$^3$; in 3D we also carry out representative simulations for an anatomically realistic bi-ventricular domain. For the simulations in the anatomically realistic human-bi-ventricular geometry, we adapt the geometry and fiber orientation for the diffusional anisotropy from Ref. [31]; the bi-ventricular geometry is enclosed in a cubical box with $512 \times 512 \times 512$ grid points; using the phase-field approach [see, e.g., Refs. [32–34]].

To trigger DADs, in the middle of the 1D cable, we introduce contiguous 30 (TP06 model) or 60 (HuVEC15 model) grid points with DAD myocytes; in 2D, we use a circular DAD-myocyte-clump (henceforth, a DAD clump) of radius 40 (TP06 model) and 80 (HuVEC15 model) grid points; in a 3D cuboid domain, we use a cylindrical DAD clump of radius 40 (TP06 model) or 80 (HuVEC15 model) and a height of 20 grid points for both the models. We model the DAD clump in human bi-ventricular geometry as an overlapping region of a DAD sphere, with a radius of 80 grid points, located in the cubical box, and the human-bi-ventricular geometry [see Fig 14(a)].

In our studies, we scale the conductances and fluxes to model the up-regulation and down-regulation of various ion channels as in Ref. [35]; e.g., to scale $G_{CaL}$, we use $G_{CaL} = S_{GCaL} \times G_{CaL0}$, where $G_{CaL0}$ is the control value for $G_{CaL}$ and $S_{GCaL}$ is the scale factor for $G_{CaL}$.

## 2.3 Ca$^{2+}$ overload protocol

DADs are transient phenomena; they occur during Ca$^{2+}$ overload in cardiac myocytes; we increase Ca entering inside the cell by scaling $G_{CaL}$ in mathematical models to induce this overload. Enhancing $I_{CaL}$ mimics the natural pathway for calcium entry, and avoids secondary ionic effects, e.g., Ca overload achieved via Na/K pump inhibition or increased $Na_i$ can introduce several secondary changes such as the changes in reversal potentials of other ionic currents (e.g., see Ref. [36]). Apart from raising the intracellular Ca, an increase in $G_{CaL}$ also increases the APD; to compensate for this increase in the APD, we increase $G_{Kr}$ by a proportional factor. The different scale factor for $G_{CaL}$ requires different $G_{Kr}$, a few representative cases are of such runs (R1-R4), for both TP06 and HuVEC15 models, are given in Table 2; for the other values of $G_{CaL}$, we use straight-line fits to get a suitable value of $G_{Kr}$ (see S1 Text).

**Table 2. The scale factors $S_{GCaL}$ and $S_{GKr}$ that we use for runs R1-R4 for the TP06 and HuVEC15 models (see text and Fig 2).**

| Sr. no. | TP06 | | HuVEC15 | |
|---|---|---|---|---|
| | $S_{GCaL}$ | $S_{GKr}$ | $S_{GCaL}$ | $S_{GKr}$ |
| R1 | 1 | 1 | 1 | 1 |
| R2 | 2 | 2.8 | 2 | 1.95 |
| R3 | 3 | 4 | 3 | 3.15 |
| R4 | 4 | 5.1 | 4 | 4.7 |

In Fig 2 we show plots of the AP (top row) for the (a) TP06 and (b) HuVEC15 models for various values of the scale factors $S_{GCaL}$ and $S_{GKr}$; these values are chosen so that the SR $Ca^{2+}$ can be increased [Figs 2 (c) and (d)] without introducing any significant changes in the APDs.

With this Ca overload, the HuVEC15 model does lead to DADs; however, the original TP06 model does not. To investigate the changes required in the model to induce DADs at the realistic Ca overload rate, we turn to Ca oscillations hypothesis proposed in Ref. [20]. Reference [20] suggests that a self sustained Ca oscillation in the Ca subsystem of a model is indicator of capability of the model to induce DADs; we use this suggestion to introduce the changes in the Ca subsystem of the TP06 model. The description of $Ca^{2+}$ subsystem contains equations for the $Na^+$-$Ca^{2+}$ exchanger, the SERCA pump, the RyR release channels, and for the $Ca^{2+}$ concentrations in various compartments. Note that the ODEs for the $V_m$ and associated ion-channel dynamics are not part of the $Ca^{2+}$ subsystem; their inclusion is not necessary and also complicates the analysis. We provide the details ODEs for the $Ca^{2+}$ subsystem for the TP06 model in S1 Text, where we carry out a continuation analysis that allows us to identify the changes required in the model to obtain DADs in the physiologically realistic regime.

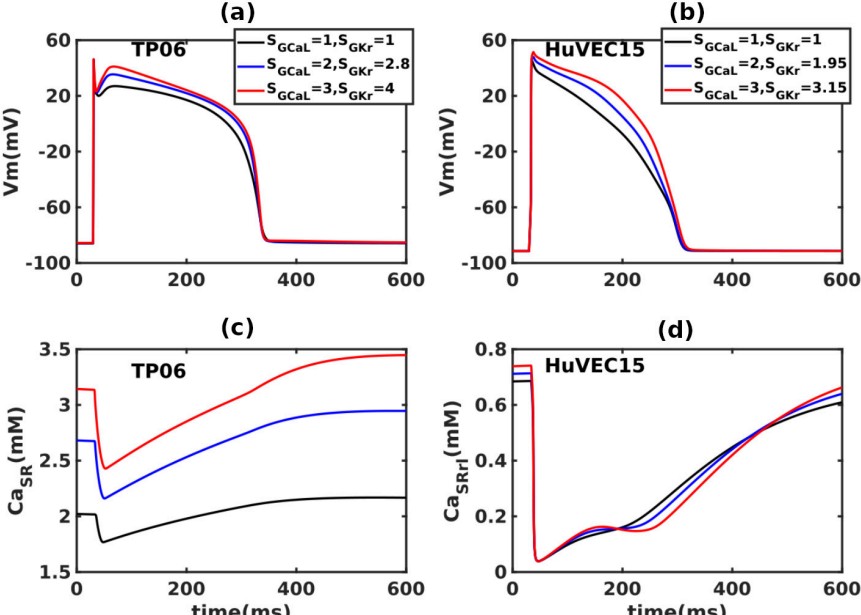

**Fig 2. The method for $Ca^{2+}$ overload:** We increase $G_{CaL}$, by changing $S_{GCaL}$, to enhance the $Ca^{2+}$ load; and we increase $G_{Kr}$, by changing $S_{GKr}$, to counterbalance the rise in the APD because of the enhancement of $G_{CaL}$. The resulting $Ca^{2+}$-overload protocol increases the Ca $^{2+}$ content in the SR; in particular, this protocol increases $[Ca^{2+}]_{SR}$ without changing the APD significantly. (a) and (b): are plots of APs for different values of $S_{GCaL}$ and $S_{GKr}$ from the TP06 and HuVEC15 models, respectively. (c) and (d): SR Calcium concentrations for the same factors of $S_{GCaL}$ and $S_{GKr}$ for both these models.

The continuation analysis reveals that introducing a RyR-mediated calcium leak from the SR to the SS is necessary to generate DADs in the TP06 model. As discussed in Sect 2.1, RyR opening in TP06 depends on calcium availability in the subspace ($Ca_{SS}$), which serves as the trigger. Without such a leak, the $Ca_{SR}$ and $Ca_{SS}$ levels required to elicit DADs are several-fold higher than the physiological range. Under normal calcium overload conditions, $Ca_{SS}$ during diastole is insufficient to induce spontaneous calcium release (SCR) and, consequently, DADs do not occur. By contrast, even a small calcium leak through closed RyRs (see Refs. [37,38]) can elevate $[Ca^{2+}]_{SS}$ during diastole to the threshold needed for SCRs and DADs. Therefore, in the TP06 model we introduce a small calcium leak representing a background SR calcium release independent of RyR opening probability through the RyR channel, as follows:

$$I_{rel} = (V_{rel}.O + V_{RyRL})([Ca^{2+}]_{SR} - [Ca^{2+}]_{SS}) \tag{5}$$

$I_{rel}$ is the molar calcium-induced calcium release (CICR) current, $O$ the opening probability of the RyR, $V_{rel} = 0.102$ ms$^{-1}$ the rate constant of the calcium release and $V_{RyRL} = 0.00018$ ms$^{-1}$ is the rate constant of calcium leak. $[Ca^{2+}]_{SR}$ and $[Ca^{2+}]_{SS}$ are the SR and subspace molar calcium concentrations, respectively. Note that, a similar RyR leak was already present in the HuVEC15 model (see S1 Text).

Without any parameter modifications, the HuVEC15 model exhibits spontaneous calcium releases (SCRs) and delayed afterdepolarizations (DADs); however, their amplitudes remain subthreshold. Given the presence of SCR, the amplitude of DADs depends on the calcium-to-voltage coupling gain (see Ref. [39]). Variations of model parameters within the physiological range did not produce suprathreshold DADs. To achieve suprathreshold DADs in the HuVEC15 model, we modified the distribution of *NCX* channels across compartments. In the original configuration, 90% of NCX channels are located in the bulk cytosol (blk), with the remaining 10% distributed within the intermediate zone (iz) adjacent to the RyRs (see Fig 1 in the main text). Notably, experimental and computational studies suggest that up to 45% of NCX channels may localize near RyRs [25]. To reflect this, we redistributed NCX channels in the model—allocating 25% near the RyRs and 75% elsewhere. This adjustment enabled the model to reproduce both supra- and subthreshold DADs within the physiological range of other parameters. In the HuVEC15 model, this distribution is controlled by the parameter $f_{NCX}$, which we also varied in this study to investigate its potential role in the initiation of DADs.

## 2.4 Parameter-sensitivity analysis

The key features in $V_m$ during DADs are: (a) the frequency of the DADs, because of multiple spontaneous calcium-ion releases; and (b) the amplitude of the DADs. We perform parameter-sensitivity analyses, as in Ref. [40], to obtain the principal model parameters that influence features (a) and (b) significantly in the TP06 and HuVEC15 models. In particular, we choose random scale factors, for the maximal conductances and the calcium fluxes, from a log-normal distribution that has a median value of 1; and we use the standard-deviation parameter $\sigma = 0.1$ to control the ranges of variation for these parameters. In this manner we generate 1000 randomly chosen factors for each one of the conductances and fluxes; we use these for the inputs into our parameter-sensitivity analysis. For the TP06 model we use 14 inputs; and for the HuVEC15 model we have 17 inputs. Next, we compute the APs for these models, for a given set of input values, by stimulating the model myocyte with a train of 500 stimuli (square current pulses of height $-52$ pA/pF and duration 1 ms for the TP06 model and height $-12$ pA/pF and duration 2.5 ms for the HuVEC15 model) with a pacing frequency of 1 Hz. For each set of randomly chosen parameter inputs, we save the last ten APs and calculate the two outputs namely the average amplitude and frequency of the DADs. To ensure consistency in the sensitivity analysis, simulations that produced suprathreshold DADs were excluded, as their presence could substantially bias the computed average DAD amplitudes and apparent pacing rate. With these outputs, we perform parameter-sensitivity analysis to obtain the parameters that influence these outputs sensitively, by constructing input and output matrices from these input and output data. With $n$, the number of samples, and $p$, the number of model parameters, we build the $n \times p$ input matrix **X**. We also construct the

$n \times m$ output matrix $\mathbf{Y}$, with $m$ the number of outputs ($m = 2$ here). By using the matrices $\mathbf{X}$ and $\mathbf{Y}$, we perform a partial-least-squares (PLS) regression to calculate the regression coefficients $\mathbf{B_{PLS}}$.

## 2.5 Ethical considerations

This computational study solely involves the analysis of existing models and no human subjects, animals, or sensitive data were directly involved in this research. Therefore, ethical approval was not required for this study.

## 3 Results

We present our results in the following sections. In Sect 3.1 we present and characterize different types of DADs in the TP06 and HuVEC15 myocyte models. Sect 3.2 deals with the results of our parameter-sensitivity analyses, which allow us to identify the parameters that affect, sensitively, the frequencies and amplitudes of the DADs in these models. We present in Sect 3.3 representative stability (or phase) diagrams, that show the regions of parameter space in which different types of DADs occur. We explore, in Sect 3.4, how the interplay of these parameters leads to such DADs. In Sect 3.5, we elaborate on a mechanism in which the NCX plays a protective role by suppressing the emergence of DADs in the TP06 model. In the final Sect 3.6 we discuss our tissue simulations, in 1D, 2D, 3D, and anatomically realistic domains into which we introduce patches with DAD myocytes.

## 3.1 Types of DADs in the TP06 and HuVEC15 models

To observe DADs in the complete TP06 and HuVEC15 models, we increase $I_{CaL}$ as we have discussed in Sect 2.3. We sampled these $V_m$ responses from a set of simulations we generated for the purpose of sensitivity analysis. For each simulation, we randomly selected scale factors for maximal conductances and calcium fluxes from a log-normal distribution, with a median of 1 and a standard deviation of 0.1. This process yielded 1000 unique combinations of scaling factors; using these scaled parameters, we obtained 1000 different $V_m$ responses; these outputs produced three types of the DADs, for which we present illustrative plots, along with a normal AP for comparison, in Fig 3. In addition to the usual sub-threshold [Figs 3 (b) and (f)] and supra-threshold [Figs 3 (d) and (h)] types of DADS, we uncover a third type, which we call multi-blip DADs [Fig 3 (c) and (g)]; these are multiple subthreshold DADs, which do not reach the activation threshold of the fast sodium-channel $I_{Na}$, between two successive APs; morphologically similar DADs have been shown in Refs. [41], and [42].

The amplitude of the DAD is defined as the peak $V_m$ of the DAD relative to the minimum potential during the diastolic interval; and its frequency is the number of DADs per second. It is important to note that multiblip DADs, occur multiple times between two successive action potentials. Additionally, the coupling interval between the preceding action potential and the first multiblip DAD is significantly shorter than that observed for single subthreshold DADs [compare Figs 3(b) and 3(c)]. In the HuVEC15 model, suprathreshold DADs were observed in conjunction with two subthreshold DADs [see Fig 3(h)]. To avoid ambiguity, any parameter set that results in a suprathreshold DAD is classified as producing suprathreshold activity, even if additional subthreshold DADs exists. We also highlight the occurrence of membrane potential oscillations during the plateau phase of the action potential, these oscillations are caused by late Ca releases (LCRs) see Fig B in S1 Text and Refs [43,44].

## 3.2 Sensitivity analysis: Crucial parameters for the DAD frequency and amplitude

We have shown different types of DADs and characterised them by using the frequency and amplitude of the DAD, $DAD_{amp}$ and $DAD_{freq}$, respectively. Here, frequency reflects the recovery of RyRs from their refractory state, a process thought to depend on the rate at which the sarcoplasmic reticulum (SR) is replenished with $Ca^{2+}$ [45].

We now perform a parameter-sensitivity analysis to determine the critical parameters for these DAD characteristics.

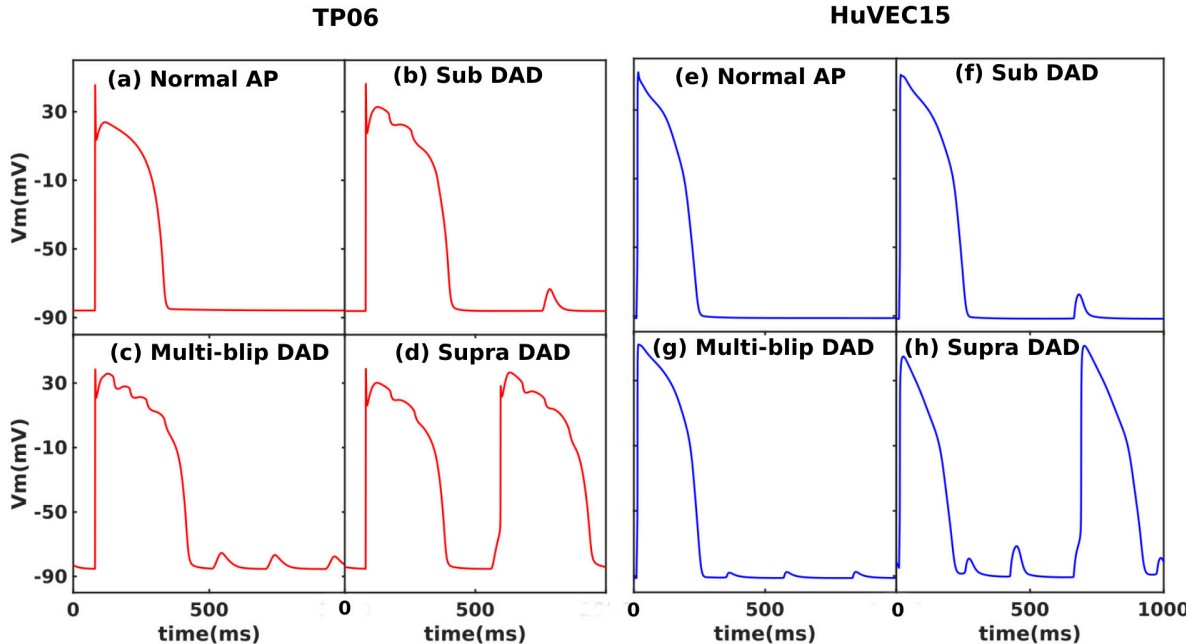

**Fig 3**. **Types of DADs:** Plots of the transmembrane potential showing the normal AP and different types of DADs (a)-(d) the TP06 model in red and (e)-(h) the HuVEC15 model in blue; (a),(e): normal AP; (b),(f): sub-threshold DADs; (c),(g): multi-blip DADs; (d),(h): supra-threshold DADs. Note that the TP06 model exhibits both DAD- and EAD-like oscillations [(b)–(d)]: the DADs are caused by spontaneous calcium releases (SCRs), whereas the EAD-like oscillations result from late calcium releases (LCRs). In contrast, the HuVEC15 model exhibits only SCRs [cf. Fig B in S1 Text].

To calculate the DAD frequency of the model, for each set of parameters, we stimulate the myocyte models for 500 pacings and record the last 10 action potentials. Near the transition point of sub-threshold DADs into the suprathreshold, a sharp change in the DAD frequency and amplitude was observed, therefore, to make the analysis unambiguous we avoided the supra-threshold DAD regime in this section. For each simulation, we calculate the average frequency and amplitude of DADs.

The parameters we choose for our sensitivity analysis are the maximal conductances of all the available transmembrane ionic currents, exchangers, the maximal SERCA pump uptake rate $V_{maxup}$, and the maximal release rate of calcium from RyRs, viz., $V_{rel}$ and the fraction of the NCX distributed in the junctional space ($f_{NCX}$).

For the DAD frequency, the analysis reveals that $G_{CaL}$ and $K_{NaCa}$ are the parameters consistent across the two models, $V_{maxup}$ is the most sensitive parameter for the TP06 model [Fig 4 (a)]; surprisingly this parameter was not most sensitive for HuVEC15 model; moreover, $f_{NCX}$ and $V_{rel}$ [see Fig 4 (c)] were the other sensitive parameter for the HuVEC15 model, but not significant for the the TP06 model. The presence of $f_{NCX}$ in the sensitivity analysis reveals that the ion-channel distribution across the various compartments are crucial for the DADs to occur. The $V_{rel}$ appears in the sensitivity analysis in HuVEC15 as it directly contributes to the RyR leak.

Note that in Figs 4 (a) and (c), the appearance of $G_{CaL}$ indicates that rate of $Ca^{2+}$ overload is a key determinant of DAD frequency. However, when the method of $Ca^{2+}$ overload changes, e.g., when it is induced by $Na_i$ accumulation the dependence on $G_{CaL}$ could disappear from the sensitivity analysis. Therefore, $G_{CaL}$ may not be a robust sensitivity parameter across different $Ca^{2+}$ overload methods.

For the DAD amplitude, in both TP06 and HuVEC15 models, we find that $K_{NaCa}$ and $G_{K1}$ are the sensitive parameters [Figs 4(b) and (d)], present across the two models. The presence of $f_{NCX}$ among the most sensitive parameters, in HuVEC15, highlights the importance of ion channel distribution across compartments in regulating DAD amplitude. The

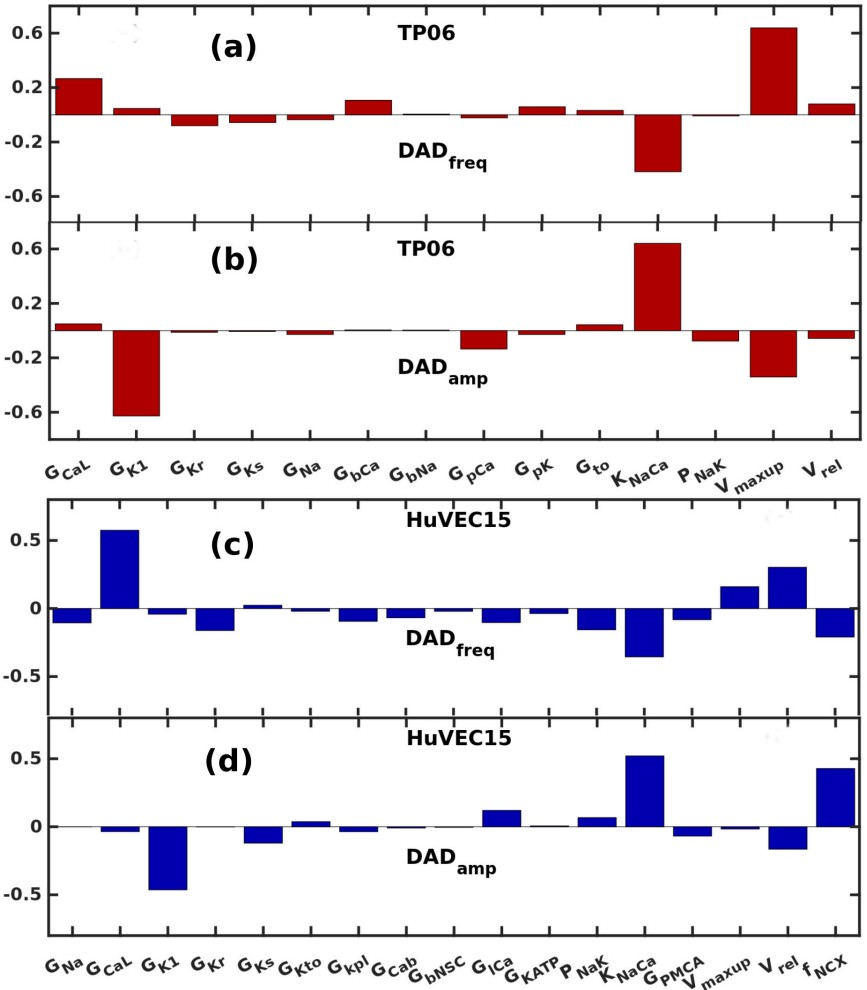

**Fig 4.** **Sensitivity analysis:** The columns of the regression-coefficient matrix $\mathbf{B_{PLS}}$ indicate how the scaling of the maximal conductances and fluxes affect $DAD_{amp}$ and $DAD_{freq}$, in red for the TP06 model and in blue for the HuVEC15 model. Sensitivity plots for: the TP06 model (a) $DAD_{freq}$ and (b) $DAD_{amp}$; the HuVEC15 model (c) $DAD_{freq}$ and (d) $DAD_{amp}$. The positive and negative values of the coefficients indicate whether an increase in the parameter increases or decreases the corresponding outputs.

role of $V_{maxup}$ in the sensitivity analysis differs between the two models: it shows the highest sensitivity in TP06 but not in HuVEC15. This difference arises because, in TP06, the SERCA pump uptake rate directly contributes to DAD generation by refilling the SR compartment, whereas in HuVEC15 the SERCA pump primarily refills the $SR_{up}$ but not the $SR_{rl}$, thereby reducing its impact in the sensitivity analysis.

The Fig 5 illustrates the quality of PLS-regression (sensitivity) predictions to the simulation. Here, we use a pacing frequency of 1 Hz. Fig G and H in S1 Text show that the results of our sensitivity analysis are robust insofar as they are not altered when we change the pacing frequency.

### 3.3 DAD phase diagrams

Now that we have determined the parameters that affect, most sensitively, DAD amplitudes and frequencies, we are in a position to present phase diagrams (or stability diagrams).

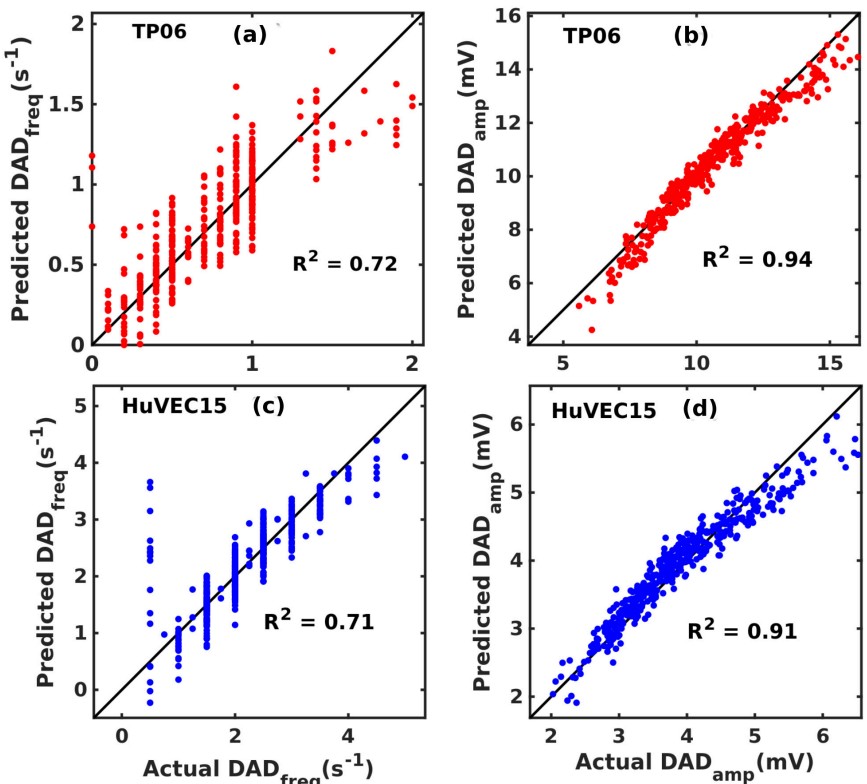

**Fig 5.** **The quality of PLS-regression predictions ($R^2$-value):** Scatter plots for the two outputs, DAD$_{freq}$ and DAD$_{amp}$, in red and blue for TP06 and HuVEC15 models, respectively, with the values computed by our numerical simulations of equations on the horizontal axis and the values estimated by the PLS regression model on the vertical axis: TP06 model (a) DAD$_{freq}$ and (b) DAD$_{amp}$; HuVEC15 model (c) DAD$_{freq}$ and (d) DAD$_{amp}$. We perform our regression analysis on a simulated data set that contains approximately 1000 samples; to obtain the value of $R^2$ value, we use 400 data points.

As there are four sensitive parameters, the full phase diagram is four-dimensional and cannot be fully represented on a 2D page (requires 4D representation). Therefore, we vary two parameters at a time while keeping the other two fixed, selecting combinations that best illustrate the phase diagram. These phase diagrams reveal the trends in DAD patterns across the parameter spaces of the TP06 and HuVEC15 models, as shown in Figs 6 (a)–(e) and Figs 7 (a)–(e), respectively.

Normal APs and those with subthreshold, multi-blip, and suprathreshold DADs are drawn, respectively, in cyan, blue, magenta, and red; the stability regions in Figs 6 and 7 follow the same color scheme. We note that, at each point in these DAD phase diagrams, for a given value of $S_{GCaL}$, we use the value of $S_{GKr}$ that is required to maintain the APD [see Figs 2 (c) and (d) and S1 Text]. There is an important difference between the DAD phase diagrams for the TP06 and HuVEC15 models: In the former, an initial increase in $S_{KNaCa}$ [see Figs 6 (a), (b), and (d)] leads to supra-threshold DADs; but, an additional increase in $S_{KNaCa}$ brings the suprathreshold-DAD phase back to the phase with normal APs. This reentry exists only up to moderate levels of $S_{GCaL}$ (i.e., $\leq 2.5$) [see Fig 6 (d)]; for a very high levels of $S_{GCaL}$ (i.e., $>2.5$), the phase diagram does not show a return from the suprathreshold DAD regime to a no-DAD regime when $S_{KNaCa}$ is increased.

This reentry exists only for the TP06 model, but not for the HuVEC15 [compare Figs 6 (a), (b), and (d) and Figs 7 (a), (b), and (d)]. The overall phase diagram can be understood as follows: for a given level of calcium overload, the occurrence of SCRs and the resulting DADs in cardiac cells happens in two steps:

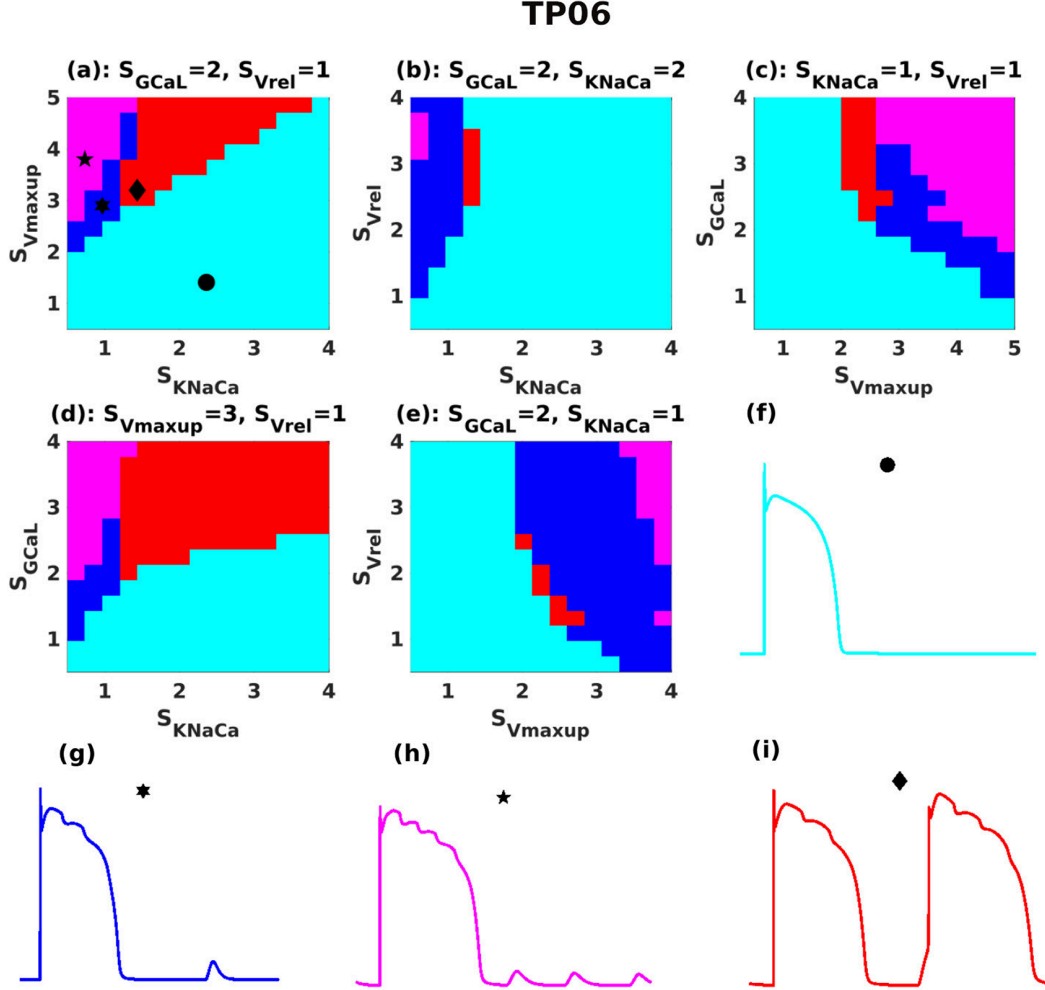

**Fig 6. TP06-model DAD phase diagrams:** (a)-(e) Phase diagrams, for various combinations of parameter values; and (f)-(i) normal APs and those with subthreshold, multi-blip, and suprathreshold DADs are drawn, respectively, in cyan, blue, magenta, and red; the stability regions in the phase diagrams follow the same color scheme.

- Calcium is spontaneously released into the cytosol during the diastolic phase.
- This calcium is then handled by two key mechanisms: the NCX (which produces a depolarizing current) and SERCA (which pumps calcium back into the sarcoplasmic reticulum). At this stage, one of two possible outcomes can occur:
  - If NCX dominates over SERCA uptake rate, this could leads to large membrane potential response, that could reach the threshold limit of an AP leading to a suprathreshold DADs.
  - If SERCA dominates over NCX, the amplitude response is small, but, at the same time, the SR fills up fast; therefore, the next abnormal Ca release from the SR, via RyR, is expected in a short time, which results in multi-blip DADs.

It is worth noting that in the TP06 model, $V_m$ oscillations occur during the plateau phase of the action potential (AP). These oscillations result from late calcium releases that induce membrane depolarizations via the NCX current. In contrast, no such oscillations are observed in the HuVEC15 model.

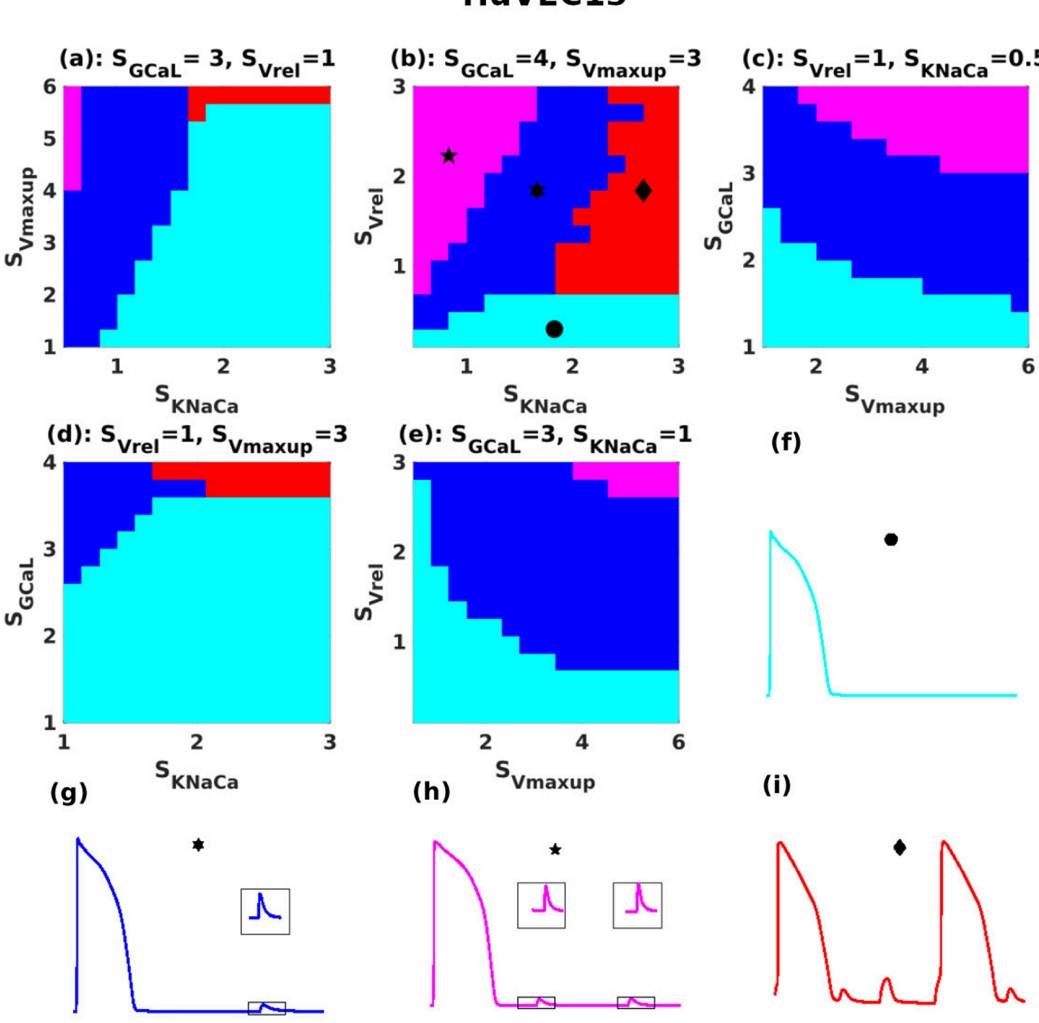

**Fig 7**. **HuVEC15-model DAD phase diagrams:** (a)-(e) Phase diagrams, for various combinations of parameter values; and (f)-(i) normal APs and those with subthreshold, multi-blip, and suprathreshold DADs are drawn, respectively, in cyan, blue, magenta, and red; the stability regions in the phase diagrams follow the same color scheme. The insets in (g) and (h) show enlarged versions of the DADs (that are highlighted with rectangles).

### 3.4 Interplay of sub-cellular Ca²⁺ components

During $Ca^{2+}$-overload, the SCRs occur via the opening of RyRs, which increase the cytosolic $Ca^{2+}$ concentrations; the NCX acts in the forward mode and extrudes excess $Ca^{2+}$ outside of the myocyte from the cytosol, whereas the SERCA pump unloads the cytosol by pumping the $Ca^{2+}$ back into the SR store. We quantify this interplay of crucial parameters of the $Ca^{2+}$-subsystem that control the frequency and amplitude of DADs. Note that, the DAD amplitudes were measured and averaged for the final 10 pacing cycles. However, if any suprathreshold DAD occurred, the maximum DAD amplitude was taken as the representative DAD amplitude. From the phase diagrams of Figs 6 (a)-(b) and Figs 7 (a)-(b), we note that suprathreshold DADs occur in the region where $S_{KNaCa}$ is large; as we reduce $S_{KNaCa}$, these systems move to regions in which subthreshold or multi-blip DADs occur.

**3.4.1 The TP06 model.** We now examine the roles of (a) the SERCA-pump uptake-rate $V_{maxup}$ and (b) the NCX control parameter $K_{NaCa}$ [see Eq S1 in S1 Text] in controlling the DAD amplitudes and frequencies in the TP06 model. In Fig 8(a) we plot the amplitude $DAD_{amp}$ versus $S_{KNaCa}$ to demonstrate that an increase in $S_{KNaCa}$ aids the subthreshold DADs to reach the threshold for triggered activity. For each value of $S_{Vmaxup}$, there is a window of values of $S_{KNaCa}$ in which we obtain suprathreshold DADs; beyond this window, an additional increase of $S_{KNaCa}$ terminates DADs; the width of this window increases with $S_{Vmaxup}$. In Fig 8(b) we plot the $DAD_{freq}$ versus $S_{KNaCa}$ to show that, initially, an increase in $S_{KNaCa}$ decreases $DAD_{freq}$; an additional increase in $S_{KNaCa}$ leads to a jump in $DAD_{freq}$, which arises from a sudden appearance of suprathreshold DADs; moreover, a very large increase in $S_{KNaCa}$ from here [see a sudden termination of the red phases in Fig 6(a) and 6(b) along the increasing direction of $S_{KNaCa}$], leads to sudden termination of DADs, i.e., we observed no DADs i.e., $DAD_{freq} = 0$.

The SERCA pump also has a critical influence on the DADs: Figs 8(c) and (d) show that non-zero $DAD_{amp}$ and $DAD_{freq}$ appear only after $S_{Vmaxup}$ crosses a threshold value. Fig 8(d) demonstrates that the increase in the $V_{maxup}$ increases the $DAD_{freq}$. $V_{maxup}$ has a small effect on $DAD_{amp}$. However, for $S_{KNaCa} = 1.44$, an increase in $S_{Vmaxup}$ leads to sudden jumps in $DAD_{amp}$ and $DAD_{freq}$; an additional increase in $S_{Vmaxup}$ reduces both of these. Thus, $S_{Vmaxup}$ and $S_{KNaCa}$ play critical roles in the formation of DADs.

**3.4.2 The HuVEC15 model.** In the HuVEC15 model, the two parameters that influence the DAD amplitude and frequency most significantly are $K_{NaCa}$ and $V_{rel}$. Our sensitivity analysis in Fig 4 shows that $K_{NaCa}$ reduces the frequency of DADs and also increases the DAD amplitude; this is confirmed by the plots of $DAD_{amp}$ and $DAD_{freq}$ versus $S_{KNaCa}$ in

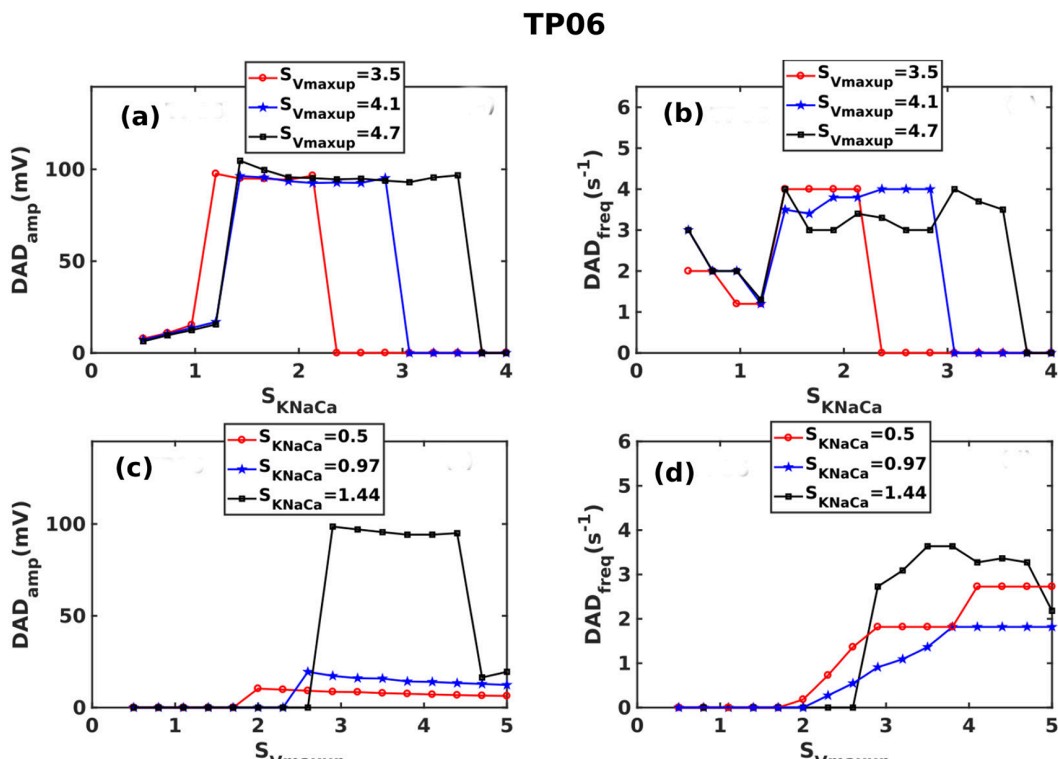

**Fig 8. Effects of the SERCA pump uptake rate $V_{maxup}$ and the NCX control parameter $K_{NaCa}$ on DADs in the TP06 model:** Plots of: $DAD_{amp}$: (a) versus $S_{KNaCa}$ for different values of $S_{Vmaxup}$ and (c) versus $S_{Vmaxup}$ for different values of $S_{KNaCa}$; $DAD_{freq}$: (b) versus $S_{KNaCa}$ for different values of $S_{Vmaxup}$ and (d) versus $S_{Vmaxup}$ for different values of $S_{KNaCa}$.

Figs 9 (a) and (b), respectively. Similarly, the plots of $DAD_{amp}$ and $DAD_{freq}$ versus $S_{Vrel}$ in Figs 9 (c) and (d), respectively, are also in consonance with our sensitivity analysis, which has demonstrated that $V_{rel}$ is the parameter that influences the frequency of DADs most sensitively and which also reduces the DAD amplitudes.

### 3.5 NCX protects against DADs in the TP06 model

The usual consensus is that $K_{NaCa}$ (NCX) enhances DAD amplitudes and is, therefore, arrhythmogenic. However, our DAD phase diagrams for the TP06 model, Figs 6(a), (b), and (d), and the plots of $DAD_{amp}$ versus $S_{KNaCa}$ [Fig 8 (a)] show that a critical value of $S_{KNaCa}$ must be reached before subthreshold DADs undergo a transition to suprathreshold DADs; but too large an increase in $S_{KNaCa}$ completely terminates the DADs. This protective mechanism is only present in the TP06 model; we do not find it in the HuVEC15 model. To elucidate this mechanism, we plot in Figs 10 (a), (b), and (c) the time series of $Ca_{SR}$, $V_m$, and the current $I_{stim}$ that provides the pacing stimuli; we compare the time series of $Ca_{SR}$ and $V_m$ for the representative values $S_{KNaCa} = 2$ and $S_{KNaCa} = 3$, after 120 pacings. The plots with the low NCX scaling factors (e.g., $S_{KNaCa} = 2$) show suprathreshold DADs; in contrast, the plots with high NCX scaling factors (e.g., $S_{KNaCa} = 3$) yield low values of $Ca_{SR}$, so DADs do not appear. Therefore, in the TP06 model, NCX reduces the $Ca_{SR}$ load because of late $Ca^{2+}$ release (LCR) (see Fig B in S1 Text), which increases the cytosolic calcium concentration during the plateau phase of the AP. This changes the direction of NCX to the forward mode, in which NCX takes one $Ca^{2+}$ ion out of the myocyte and puts three $Na^+$ ions back into the myocyte. We recall that, in the backward or reverse mode, NCX expels three $Na^+$ ions from the myocyte in exchange for one $Ca^{2+}$ ion; during the plateau region of the AP, with the normal physiological

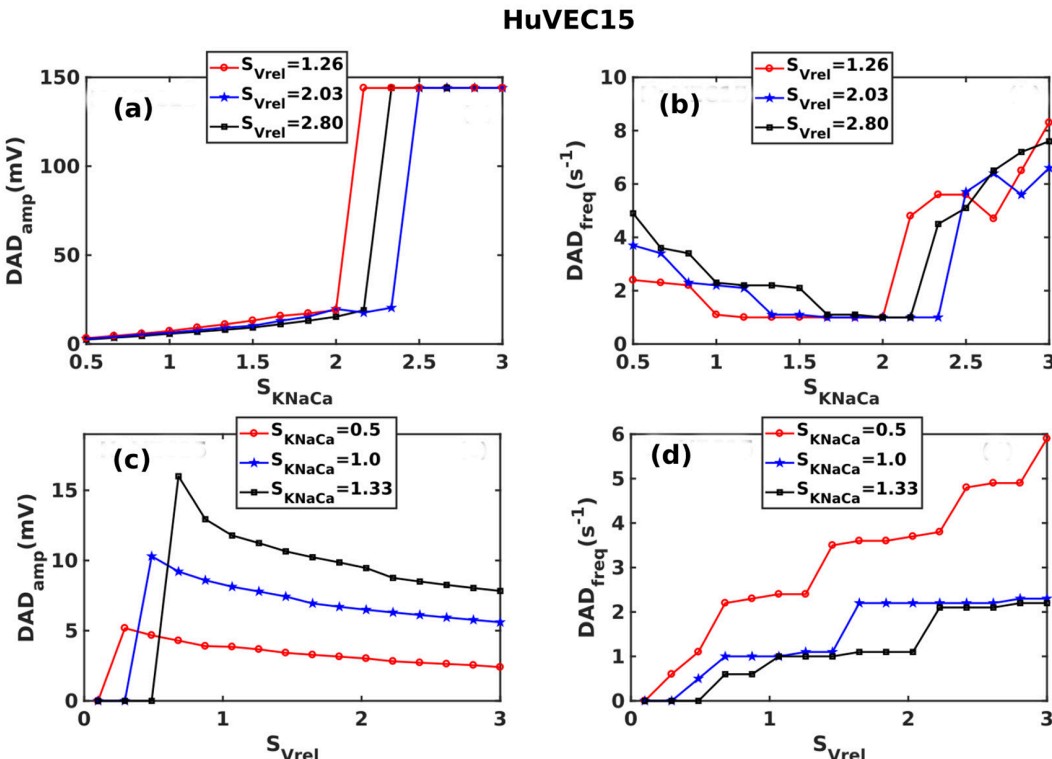

**Fig 9**. **The Role of RyR release rate $V_{rel}$ and $K_{NaCa}$ on DADs in the HuVEC15 model:** Plots of: $DAD_{amp}$: (a) versus $S_{KNaCa}$ for different values of $S_{Vrel}$ and (c) versus $S_{Vrel}$ for different values of $S_{KNaCa}$; $DAD_{freq}$: (b) versus $S_{KNaCa}$ for different values of $S_{Vrel}$ and (d) versus $S_{Vrel}$ for different values of $S_{KNaCa}$.

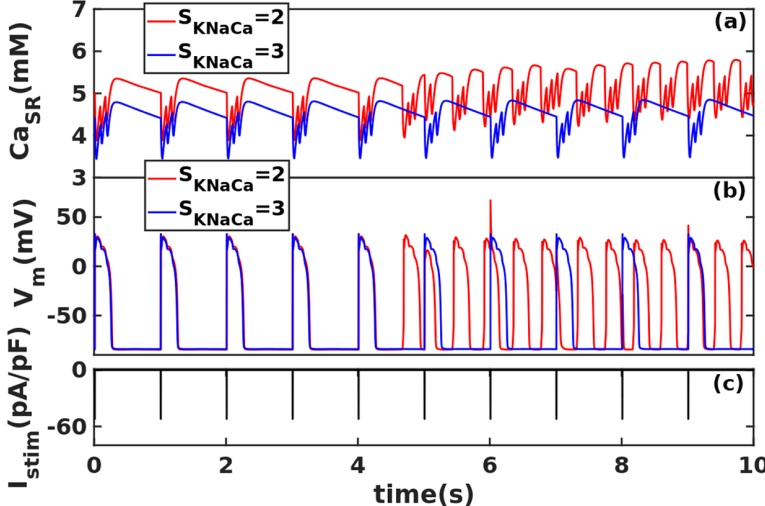

**Fig 10. The Role of $K_{NaCa}$ in terminating DADs in TP06 model:** Plots versus time $t$ of (a) $Ca_{SR}$, (b) $V_m$, and (c) the current $I_{stim}$ that provides the pacing stimuli, for two different values of $S_{KNaCa}$ [with $S_{GCaL} = 2$, $S_{GKr} = 2.8$, $S_{Vrel} = 1$ and $S_{Vmaxup} = 4$]. In (a) $Ca_{SR}$ is lower for $S_{KNaCa} = 3$ than for $S_{KNaCa} = 2$; in (b) suprathreshold DADs are triggered if $S_{KNaCa} = 2$, but are absent if $S_{KNaCa} = 3$.

value of Na$^+$, NCX operates in the reverse mode; a sudden increase in cytosolic Ca$^{2+}$ forces the NCX to operate in the forward mode. Such LCRs do not occur in the HuVEC15 model, so NCX does not protect against DADs in this model. This is related to the important difference [cf. Sect 3.3] between the DAD phase diagrams of the TP06 and HuVEC15 models. We give a representative plot for the direction of NCX and SCRs (Fig B in S1 Text).

### 3.6 Cable and tissue simulations

We have presented our results for isolated myocytes; we now describe our results from representative simulations, for both TP06 and HuVEC15 models, in cable and tissue domains with DAD clumps [see Sect 2.2]. In these domains, myocytes are electrotonically coupled. When DADs occur in the clump, $V_m$ rises above its value in the resting state, and the electrotonic currents start flowing to the other resting myocytes. The competition between the rising DAD of a certain duration and diffusion processes in cardiac-tissue models leads to the emergence of a length scale [46], which depends on the parameters of the model; if the linear size of the DAD clump exceeds this length scale, then the clump of DAD myocytes can fire focal excitations. In a DAD clump, myocytes are synchronized by the pacings; therefore, they fire synchronously. We illustrate this in our tissue simulations.

In Fig 11 we present pseudocolor space-time plots of $V_m$ in a cable with DAD clumps for both the TP06 (panel (i)) and HuVEC15 (panel (ii)) models; in subfigures (a), (b), (c), and (d) the DAD clumps have a normal AP, subthreshold DADs, multi-blip DADs, and suprathreshold DADs, respectively [cf. Figs 6 and 7]. We use a 1Hz current stimulus for the first myocyte in the cable and track the signal as it propagates to the other end.

Fig 11(a) shows the propagation of a normal signal from the first myocyte to the last myocyte of the cable; Fig 11(b) displays the emergence of sub-threshold depolarization at the center of the cable, following a normal excitation; Fig 11(c) exhibits a series of subthreshold depolarizations (multi-blips) at the center of the cable, following a normal excitation; and Fig 11(d) depicts the emergence of PVCs, following a normal excitation. We find that, in these cable domains, DAD clumps with 30 and 60 grid points representing suprathreshold myocytes are sufficient for triggering PVCs in the TP06 and HuVEC15 models, respectively.

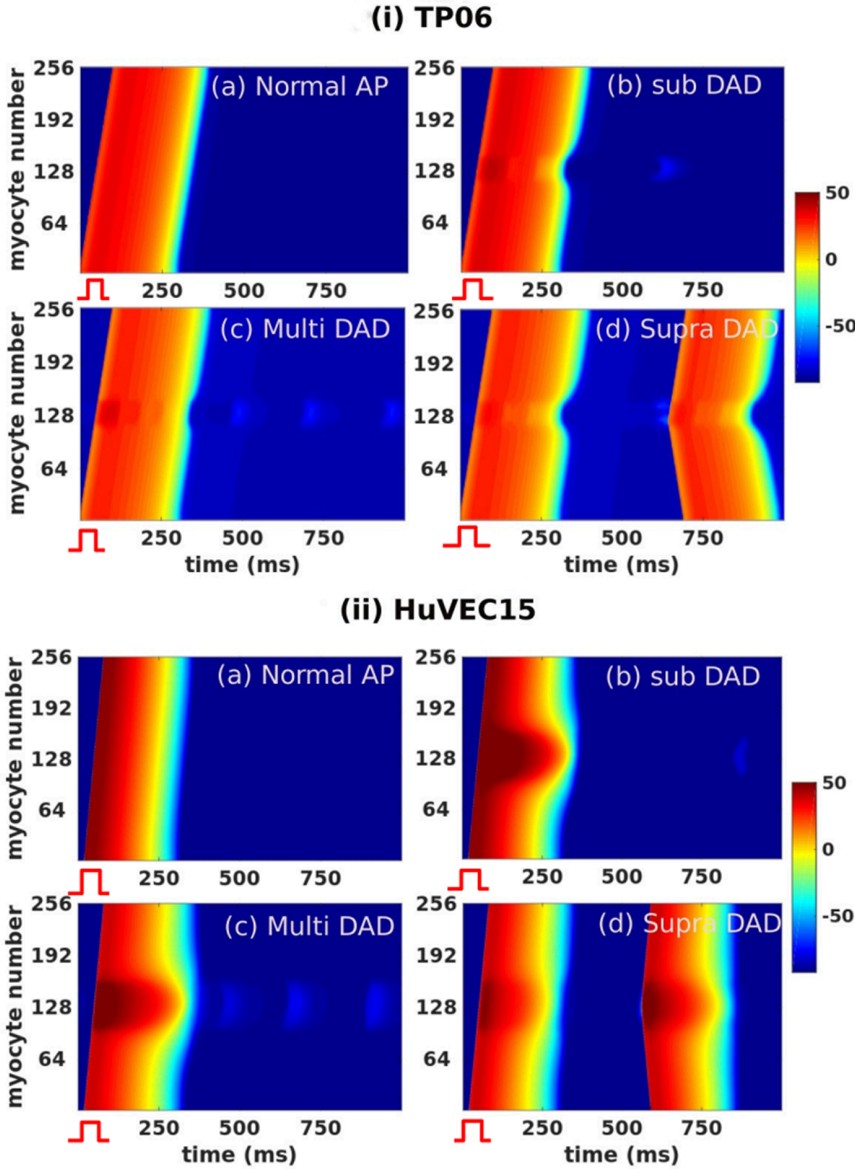

**Fig 11. Cable simulations:** Pseudocolor space-time plots of $V_m$(mV) along our 1D cable domain; DAD myocytes occupy the middle region of the cable (30 and 60 grid points for the TP06 and HuVEC15 models, respectively). The subplots (i) and (ii) show: (a) a normal AP; (b) subthreshold DADs; (c) multi-blip DADs; and (d) suprathreshold DADs. The stimulation frequency we choose is 1 Hz and the stimulus is applied to the first grid point.

In Fig 12 we present pseudocolor plots of $V_m$ from our simulations for 2D domains for the TP06 (panel (i)) and HuVEC15 (panel (ii)) models. These plots illustrate the effects of circular DAD clumps [see Sect 2.2] on the propagation of plane waves of electrical activation through these domains, as we pace the tissue at its left boundary. If the clump comprises myocytes with normal APs, then we find normal excitation propagation [subfigures (a)]; if the clump consists of subthreshold DAD myocytes, we observe the emergence of sub-threshold excitation in [subfigures (b)]; for a clump with multi-blip DAD myocytes, we find multiple subthreshold excitations, with the clump itself containing the subthreshold DADs; with a clump of supra-threshold DAD myocytes, PVCs emerge after 6 pacings and then propagate through the entire domain.

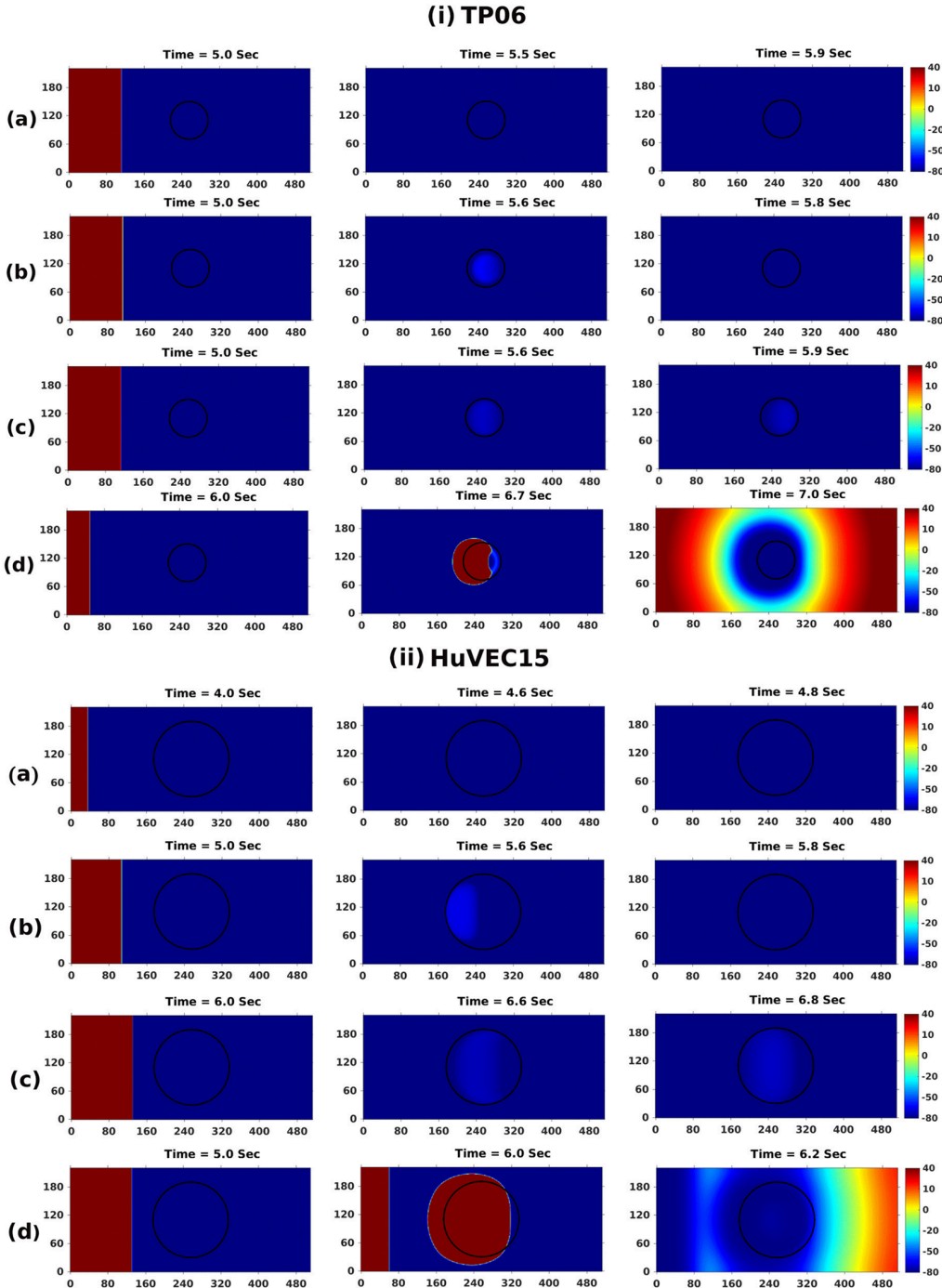

**Fig 12. 2D simulations:** Pseudocolor plots of $V_m$(mV) for the TP06 (panel (i)) and HuVEC15 (panel (ii)) models illustrating the effects of pacing on DAD clumps in cardiac tissue: (a) myocytes with normal APs (no DADs observed); (b) subthreshold DAD myocytes (after a few pacings the clump fires a PVC, which does not reach the threshold for full excitation and, therefore, is localised in a limited region); (c) multi-blip DAD myocytes (multiple PVCs emerge, but they do not reach the full excitation threshold and, therefore, are localised in a limited region); (d) suprathreshold DAD myocytes (a full-strength PVC emerges and excites the entire domain). For the complete spatiotemporal evolution of $V_m$ see Videos [S1-S4] (for the TP06 model) and Videos [S5-S8] (for the HuVEC15 model) in the Supporting information.

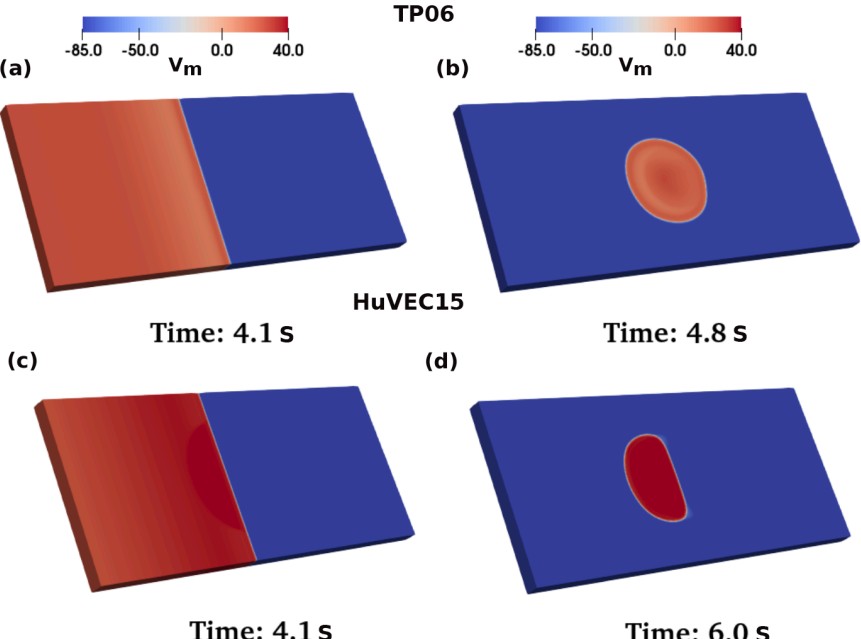

**Fig 13. 3D slabs:** Pseudocolor plots of $V_m$(mV), for the TP06 and HuVEC15 models, illustrating the spatiotemporal evolution of electrical excitation and the emergence of PVCs from a DAD-myocyte clump, with suprathreshold DADs. TP06 model: (a) pacing-induced plane-wave propagation; (b) PVC emerging after 4 pacings. HuVEC15 model: (c) pacing-induced plane-wave propagation; (d): PVC emerging after 5 pacings. For the complete spatiotemporal evolution of $V_m$, see Video S9 (for the TP06 model) and Video S10 (for the HuVEC15 model)] in the Supporting information.

In Fig 13 we present representative simulations in 3D square-cuboid domains, for the TP06 and HuVEC15 cardiac-tissue models, with a cylindrical DAD clump. Our results are qualitatively similar to those we have presented in 2D. We find, e.g., that PVCs emerge from this clump after 4 and 6 pacings, respectively, in the TP06 and HuVEC15 models.

In our most realistic study, we perform a full-heart simulation by using the phase-field method; we include fiber orientation to account for the anisotropy of cardiac tissue (see, e.g., Refs. [31,33]). We include a DAD clump as we have described in Sect 2.2. In Fig 14(a), we show such a suprathreshold DAD clump for the TP06 model embedded in a human-bi-ventricular geometry, with roughly 775,000 grid points. In Fig 14(b), we present a pseudocolor plot of the $V_m$ in this geometry; this depicts the propagation of a normal stimulus, applied at the apex of the human bi-ventricular geometry. Once this propagating wave of electrical activation encounters the DAD clump, PVCs emerge, as we show in Fig 14(c): these PVCs propagate and finally excite both the ventricles as we can see in Fig 14(d).

## 4 Discussion and conclusions

We have carried out a detailed investigation of DADs in two human-ventricular myocyte models, the TP06 and HuVEC15, and we have compared the DADs in these models at both single-cell and tissue levels.

By increasing $G_{CaL}$ in the full myocyte models, we have obtained calcium-overload conditions and have shown the following: under Ca$^{2+}$ overload, the TP06 model shows late Ca releases (LCRs), which lead to EAD-type depolarizations without reopening the $I_{CaL}$ channel (see Fig B in S1 Text); this LCR driven EAD compromises the APD balance between $I_{CaL}$ and $I_{Kr}$ that we implement in the beginning. This increase in APD due to LCRs is not unique to our study; this has also been reported in Refs. [39,43,44,47,48]; however, the HuVEC15 model did not exhibit these LCRs.

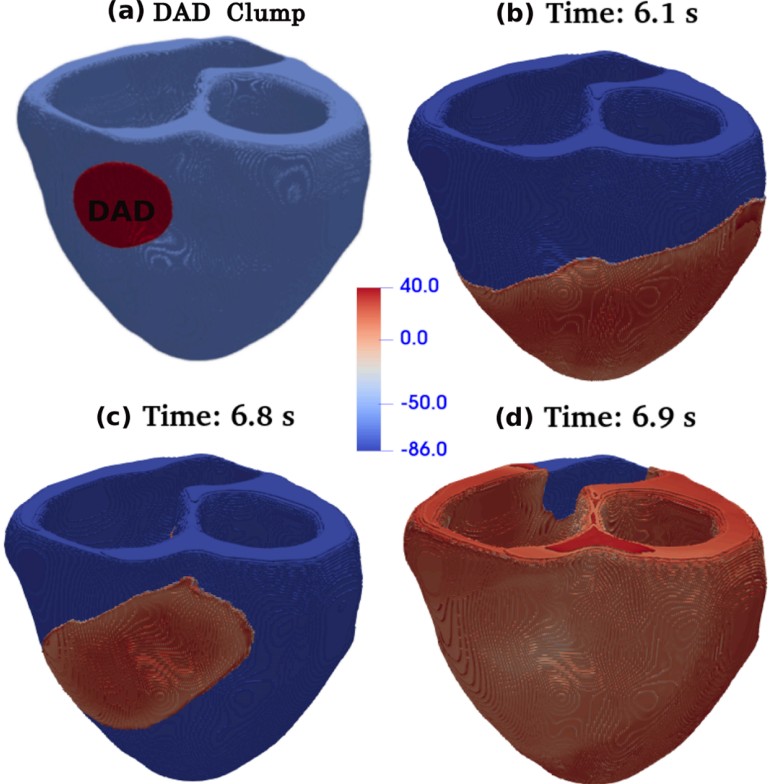

**Fig 14. Biventricular simulations:** (a) Pseudocolor plot showing a representative clump of DAD myocytes (red) embedded in a human biventricular geometry (blue). Pseudocolor plots of $V_m$(mV) depicting (b) normal excitation propagation after stimulation of the bi-ventricular geometry from its apex, (c) emergence of PVCs from the DAD clump after 6 pacings (1 Hz pacing frequency), and (d) the propagation of PVCs and subsequent scroll-wave excitations to both the ventricles. For the complete spatiotemporal evolution of $V_m$ see Video S11 in the Supporting information.

By incorporating leaky RyR channels (in the TP06), and modifying the spatial distribution of NCX channels (in HuVEC15), we successfully induced various types of DADs in both the TP06 and HuVEC15 models. Through systematic variation of key parameters—including $G_{CaL}$, $V_{rel}$, $K_{NaCa}$, and $V_{maxup}$—we identified three distinct types of DADs: (a) subthreshold, (b) suprathreshold, and (c) multi-blip DADs, the latter characterized by multiple subthreshold events occurring between two consecutive action potentials. To differentiate among these DAD types, we introduced two defining metrics: DAD amplitude ($DAD_{amp}$) and DAD frequency ($DAD_{freq}$), and analyzed their sensitivity to the underlying model parameters.

Our parameter-sensitivity analysis reveals that, in the TP06 model, the most significant contributors to DAD amplitude ($DAD_{amp}$) are $K_{NaCa}$, $G_{K1}$, and $V_{maxup}$. In the HuVEC15 model, the key parameters are $K_{NaCa}$, $f_{NCX}$, and $G_{K1}$. These findings are consistent with previous studies [49–53]. Notably, in the TP06 model, $V_{maxup}$ exhibits a negative influence on $DAD_{amp}$, as it enhances calcium reuptake into the sarcoplasmic reticulum, thereby reducing cytosolic calcium levels. This action counteracts the electrogenic activity of the sodium-calcium exchanger ($K_{NaCa}$), ultimately limiting the rise in DAD amplitude. In the HuVEC15 model, $f_{NCX}$ denotes the fraction of sodium-calcium exchangers localized near the intermediate zone (iz, see Fig 1), a region with elevated calcium concentration relative to the bulk cytosol; as a result, higher values of $f_{NCX}$ contribute positively to $DAD_{amp}$ through increased electrogenic exchange current.

Our analysis demonstrates that DAD frequency ($DAD_{freq}$) is primarily influenced by $G_{CaL}$, $K_{NaCa}$, and $V_{maxup}$ in the TP06 model, and by $G_{CaL}$, $K_{NaCa}$, $f_{NCX}$, and $V_{rel}$ in the HuVEC15 model. The sensitivity of $DAD_{freq}$ to these parameters can be

understood as follows: (a) $G_{CaL}$ contributes to intracellular calcium overload, a necessary condition for DAD generation; (b) $K_{NaCa}$ and $f_{NCX}$ (specific to the HuVEC15 model) exert a negative effect on $DAD_{freq}$, as they promote calcium extrusion via forward-mode NCX activity; (c) an increase in $V_{maxup}$ enhances sarcoplasmic reticulum (SR) calcium reuptake, thereby facilitating subsequent calcium release events; and (d) in the HuVEC15 model, $V_{rel}$ controls the SR calcium leak in the junctional space and therefore increasing the chances of DADs and $DAD_{freq}$.

Guided by the insights obtained from our sensitivity analysis, we systematically varied key parameters to construct phase diagrams that capture the dynamic behavior of DADs. In each diagram, we varied two of the most influential parameters while holding the remaining two constant. This approach revealed several noteworthy trends. For example, at elevated calcium loading rates (increased $G_{CaL}$), both suprathreshold and multi-blip DADs are more likely to occur, with their prevalence depending on the calcium uptake rate via the SERCA pump ($V_{maxup}$); see Fig 6(c). Another important observation is the role of $K_{NaCa}$ in modulating DAD behavior. While increasing $K_{NaCa}$ generally enhances DAD amplitude and promotes suprathreshold DADs, we identified a threshold in the TP06 model beyond which further increases in $K_{NaCa}$ lead to the suppression of suprathreshold DADs. This effect does not occur in the HuVEC15 model and appears to be linked to the presence of late calcium releases (LCRs), which are present in the TP06 model but absent in HuVEC15. In TP06, LCRs raise cytosolic calcium levels during the late phase of the action potential, thereby activating NCX in the forward mode. A sufficiently large $K_{NaCa}$ enhances calcium extrusion under these conditions, potentially eliminating spontaneous calcium releases (SCRs) and the resulting DADs. A similar suppression of DADs at high $K_{NaCa}$ values has been reported in Ref. [54], though without accounting for LCRs. Our findings suggest that the presence of LCRs may further reinforce this termination mechanism.

Although the calcium handling formulations differ between the TP06 and HuVEC15 models, our results consistently identify key factors that promote spontaneous calcium releases (SCRs) and delayed afterdepolarizations (DADs) in both. Previous studies have reported conflicting findings regarding the role of the SERCA pump. For instance, Refs. [55,56] suggest that increasing SERCA uptake rate enhances DAD incidence, in agreement with our findings. In contrast, other studies argue that SERCA uptake rate activity suppresses DADs [20,57]. These discrepancies may arise from differences in (a) the sensitivity of RyR activation to calcium concentrations in the SR or subspace, or (b) the methods used to induce calcium overload. For example, Ref. [20] induces calcium overload via intracellular sodium accumulation ($Na_i$), whereas our approach involves increasing $I_{CaL}$.

As discussed in Sect 2.1, both the TP06 and HuVEC15 models lack explicit RyR inactivation mechanisms. However, the HuVEC15 model incorporates a more compartmentalized structure than TP06, leading to several important consequences. First, in the TP06 model, the absence of RyR inactivation and lack of compartmentalization enables the occurrence of late calcium releases (LCRs) during the plateau phase of the action potentials. These LCRs give rise to subthreshold, early-afterdepolarization (EAD)-like membrane potential ($V_m$) oscillations. Interestingly, this behavior is not observed in the HuVEC15 model. We conjecture that this difference arises due to the distinct SR architecture in HuVEC15: a substantial calcium release from the SR release compartment ($SR_{rl}$, see schematic in Fig 1) may rapidly deplete its content, thereby preventing subsequent release during the plateau phase of the same AP [24]. The refilling of $SR_{rl}$ in HuVEC15 depends on diffusion from the SR uptake compartment ($SR_{up}$), which introduces a rate-limiting step. A second consequence of this diffusion-limited refilling is reflected in the progression from isolated subthreshold DADs to increasing DAD amplitudes and eventually to suprathreshold DADs [see Fig 1(h)]. Notably, we observe multi-blip DADs in both TP06 and HuVEC15 models. This suggests that subthreshold DADs may not deplete SR calcium stores fully to prevent subsequent release events.

The multi-blip DADs are similar to diastolic-membrane-potential oscillations reported in the sinoatrial node (SAN) [42], a Purkinje-cell model [58], and in myocardial myocytes [41].

Our findings highlight the potential pro-arrhythmic impact of subthreshold and multi-blip DADs at the cellular and tissue levels. Subthreshold DADs are known to inactivate sodium channels ($I_{Na}$) [59], and our results confirm that both subthreshold and multi-blip DADs reduce sodium channel availability [Fig F in S1 Text]. Importantly, multi-blip DADs lead to repeated inactivation of $I_{Na}$, which could increase the risk of conduction block (see Ref. [60]).

Moreover, a reduction in $G_{K1}$ facilitates the progression of subthreshold and multi-blip DADs into suprathreshold events [see Fig I in S1 Text]. Given that multi-blip DADs typically occur with short coupling intervals, this transition increases the likelihood of a DAD driven PVCs with a short coupling interval; a case of short coupled PVCs leading to VT and VF have been reported in Ref. [61], however, the subcellular cause of these PVCs is unclear.

In Sect 3.6, we analyzed the behavior of DAD clumps comprising the three types of DAD-generating myocytes, using both the TP06 and HuVEC15 models. Our investigations spanned multiple spatial scales, including 1D cables, 2D tissue sheets, and a 3D anatomically realistic human bi-ventricular geometry. When the linear dimension of a DAD clump exceeds a threshold—dependent on specific model parameters—it can generate PVCs. For a comprehensive discussion on the critical clump size required to initiate PVCs, we refer the reader to Ref. [46]. Notably, our simulations show that a normal pacing stimulus applied at the ventricular apex propagates through the tissue and interacts with the DAD clump, which then becomes a source of PVCs [Figs 14(c) and (d)]. These PVCs can be self-sustaining, i.e. continuing even in the absence of external pacing. Similar forms of sustained triggered activity have been observed experimentally at the isolated myocyte level [9].

Thus, we have shown that both TP06 and HuVEC15 models are useful for studying PVCs, induced by different types of DAD clumps in tissue and bi-ventricular domains. The evolution of PVCs, at the organ scale, and $Ca^{2+}$ dynamics, at the subcellular scale, are bidirectionally coupled; for a detailed discussion see Ref. [62]. During calcium overload, LCRs, which frequently accompany diastolic SCRs and DADs, can play a crucial role in the dynamics of $V_m$ and $Ca^{2+}$ overload (see Ref. [63]). We have shown that both LCRs and SCRs can occur in the TP06 model [Figs 3(b)-(d) and, in S1 Text, Fig B]. Therefore, the TP06 model can be a candidate for the examination of arrhythmias that involves both the SCRs and LCRs together; such LCRs were observed in the HuVEC15 model as well (data not shown), but only when we do not maintain the APD, when increasing the $I_{CaL}$.

Thus, we were able to demonstrate a range of dynamics in both models under different parameter regimes; moreover, we illustrate that taking into account the Ca compartmentalization could significantly change the DAD outcome in the models, therefore, will be crucial to include in the future studies. In future it would be interesting to further investigate the mechanism for EAD- and DAD-driven cardiac arrhythmias using these models.

## 5 Limitations of our study

The origin of DADs is related to the sub-cellular phenomena of calcium sparks and calcium waves; our study does not discuss the latter in detail. The duration, width, and the steepness of the rise of DADs are not captured accurately in the common-pool models of the type we use here; these factors influence the critical-DAD-clump size requirement for the emergence of PVCs; therefore, we have not addressed this requirement here. The critical-clump size requirements are better captured in the models with an accurate description of SCR waveforms. Moreover, the stochasticity in the calcium sparks is valuable to understand the inherent randomness at the cellular level, which is not addressed by these common pool models; but the common pool models provides a complementary perspective for investigating DAD-mediated arrhythmia mechanisms. The common pool models offer easy and more efficient implementation in terms of computational efficiency, that could be helpful in linking cellular and tissue-scale phenomena. We will address randomness in our future studies.

## Supporting information

**S1 Video.** Animation of the pseudocolor plots of $V_m$(mV), for the TP06 model, illustrating the spatiotemporal evolution of plane wave pacing in cardiac tissue as in Fig 12(i)(a). The parameter set we use is: $S_{GCaL} = 1.0$, $S_{Vmaxup} = 1.0$, $S_{KNaCa} = 1.0$, $S_{Vrel} = 1.0$, $S_{GKr} = 1.0$ . For the video, we use 30 frames per second with each frame separated from the succeeding frame by 20ms in real-time.
(MP4)

**S2 Video.** Animation of the pseudocolor plots of $V_m$(mV), for the TP06 model, illustrating the spatiotemporal evolution of plane-wave pacing in cardiac tissue and the emergence of PVCs, from the subthreshold-DAD clump, as in Fig 12(i)(b). The parameter set we use is: $S_{GCaL} = 2.0$, $S_{Vmaxup} = 3.0$, $S_{KNaCa} = 1.0$, $S_{Vrel} = 1.0$, $S_{GKr} = 1.0$. For the video, we use 30 frames per second (fps), with an inter-frame separation (ifs) of 20 ms in real-time.
(MP4)

**S3 Video.** Animation of the pseudocolor plots of $V_m$(mV), for the TP06 model, illustrating the spatiotemporal evolution of plane-wave pacing in cardiac tissue and the emergence of PVCs, from the multiblip-DAD clump, as in Fig 12(i)(c). The parameter set we use is: $S_{GCaL} = 2.0$, $S_{Vmaxup} = 4.5$, $S_{KNaCa} = 0.8$, $S_{Vrel} = 1.0$, $S_{GKr} = 1.0$. For the video, we use fps= 30 and ifs= 20ms in real-time.
(MP4)

**S4 Video.** Animation of the pseudocolor plots of $V_m$(mV), for the TP06 model, illustrating the spatiotemporal evolution of plane-wave pacing in cardiac tissue and the emergence of PVCs, from the suprathreshold-DAD clump, as in Fig 12(i)(d). The parameter set we use is: $S_{GCaL} = 2.0$, $S_{Vmaxup} = 3.0$, $S_{KNaCa} = 2.5$, $S_{Vrel} = 1.0$, $S_{GKr} = 1.0$. For the video, we use fps = 30 and ifs = 20ms in real-time.
(MP4)

**S5 Video.** Animation of the pseudocolor plots of $V_m$(mV), for the HuVEC15 model, illustrating the spatiotemporal evolution of plane-wave pacing in cardiac tissue, as in Fig 12(ii)(a). The parameter set we use is: $S_{GCaL} = 1.0$, $S_{Vmaxup} = 1.0$, $S_{KNaCa} = 1.0$, $S_{Vrel} = 1.0$, $S_{GKr} = 1.0$. For the video, we use fps = 30 and ifs = 20ms in real-time.
(MP4)

**S6 Video.** Animation of the pseudocolor plots of $V_m$(mV), for the HuVEC15 model, illustrating the spatiotemporal evolution of plane-wave pacing in cardiac tissue and the emergence of PVCs, from the subthreshold-DAD clump, as in Fig 12(ii)(b). The parameter set we use is: $S_{GCaL} = 4.0$, $S_{Vmaxup} = 3.0$, $S_{KNaCa} = 1.0$, $S_{Vrel} = 1.2$, $S_{GKr} = 4.7$, $S_{GK1} = 1$. For the video, we use fps = 30 frames per second with each frame separated from the succeeding frame by ifs = 20ms in real-time.
(MP4)

**S7 Video.** Animation of the pseudocolor plots of $V_m$(mV), for the HuVEC15 model, illustrating the spatiotemporal evolution of plane-wave pacing in cardiac tissue and the emergence of subthreshold PVCs multiple times, from the multiblib-DAD clump, as in Fig 12(ii)(c). The parameter set we use is: $S_{GCaL} = 4.0$, $S_{Vmaxup} = 3.0$, $S_{KNaCa} = .7$, $S_{Vrel} = 2.2$, $S_{GKr} = 4.7$, $S_{GK1} = 0.35$. For the video, we use fps = 30 and ifs = 20ms in real-time.
(MP4)

**S8 Video.** Animation of the pseudocolor plots of $V_m$(mV), for the HuVEC15 model, illustrating the spatiotemporal evolution of plane-wave pacing in cardiac tissue and the emergence of suprathreshold PVCs, from the DAD clump, as in

Fig 12(ii)(d). The parameter set we use is: $S_{GCaL} = 4.0$, $S_{Vmaxup} = 3.0$, $S_{KNaCa} = 1.0$, $S_{Vrel} = 1.0$, $S_{GKr} = 5.2$, $S_{GK1} = 1.0$. For the video, we use fps = 30 and ifs = 20ms in real-time. For the video, we use fps = 30 and ifs = 20ms in real-time. (MP4)

**S9 Video.** Animation of the pseudocolor plots of $V_m$(mV), for the TP06 model, illustrating the spatiotemporal evolution of plane-wave pacing in cuboidal cardiac tissue, with a disc-shaped suprathreshold-DAD clump embedded in it, and the emergence of PVCs from the clump, as in Fig 13(a)-(b). The parameter set we use is: $S_{GCaL} = 2.5$, $S_{Vmaxup} = 4.5$, $S_{KNaCa} = 2.5$, $S_{Vrel} = 1.0$, $S_{GKr} = 2.8$. For the video, we use fps = 30 and ifs = 20ms in real-time. (MP4)

**S10 Video.** Animation of the pseudocolor plots of $V_m$(mV), for the HuVEC15 model, illustrating the spatiotemporal evolution of plane-wave pacing in cuboidal cardiac tissue, with a disc-shaped suprathreshold-DAD clump embedded in it, and the emergence of PVCs from the clump, as in Fig 13(c)-(d). The parameter set we use is: $S_{GCaL} = 3.0$, $S_{Vmaxup} = 3.0$, $S_{KNaCa} = 2.7$, $S_{Vrel} = 1.0$, $S_{GKr} = 5.2$, $S_{GK1} = 1.0$. For the video, we use fps = 30 and ifs = 20ms in real-time. (MP4)

**S11 Video.** Animation of the pseudocolor plots of $V_m$(mV), for the TP06 model, illustrating the spatiotemporal evolution of $V_m$(mV) in a human bi-ventricular geometry, with a suprathreshold-DAD clump embedded in it, and the emergence of PVCs from the clump, as in Fig 14(b)-(d). The parameter set we use is: $S_{GCaL} = 2.5$, $S_{Vmaxup} = 4.5$, $S_{KNaCa} = 2.5$, $S_{Vrel} = 1.0$, $S_{GKr} = 2.8$. For the video, we use fps = 30 and ifs = 20ms in real-time. (MP4)

**S1 Text.** In this document we present the extended methods, results and analysis. (PDF)

## Acknowledgment

We thank Mahesh K. Mulimani and Soling Zimik for valuable discussions.

## Author contributions

**Conceptualization:** Navneet Roshan.

**Data curation:** Navneet Roshan.

**Formal analysis:** Navneet Roshan.

**Funding acquisition:** Rahul Pandit.

**Methodology:** Navneet Roshan.

**Project administration:** Rahul Pandit.

**Resources:** Rahul Pandit.

**Software:** Navneet Roshan.

**Supervision:** Rahul Pandit.

**Validation:** Navneet Roshan.

**Writing – original draft:** Navneet Roshan, Rahul Pandit.

**Writing – review & editing:** Navneet Roshan, Rahul Pandit.

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
