## [Decision Letter · Decision Letter 0]

7 Oct 2024

PONE-D-24-32626Abnormal Calcium Release and Afterdepolarizations: A Comparison of Two Mathematical Models for Human Ventricular MyocytesPLOS ONE

Dear Dr. Roshan,

Thank you for submitting your manuscript to PLOS ONE. After careful consideration, we feel that it has merit but does not fully meet PLOS ONE’s publication criteria as it currently stands. Therefore, we invite you to submit a revised version of the manuscript that addresses the points raised during the review process.

We look forward to receiving your revised manuscript.

Kind regards,

Daniel M. Johnson, PhD

Academic Editor

PLOS ONE

Journal Requirements:

4. In the online submission form, you indicated that “All the data will be made avaialbe upon request”. All PLOS journals now require all data underlying the findings described in their manuscript to be freely available to other researchers, either 1. In a public repository, 2. Within the manuscript itself, or 3. Uploaded as supplementary information. This policy applies to all data except where public deposition would breach compliance with the protocol approved by your research ethics board. If your data cannot be made publicly available for ethical or legal reasons (e.g., public availability would compromise patient privacy), please explain your reasons on resubmission and your exemption request will be escalated for approval.

5. Please upload a copy of Supporting Information which you refer to in your text on page 14.

Additional Editor Comments:

All Reviewers commented on the fact that the aims and rationale of the manuscript need to be better defined, and that there needs to be an improved positioning of the manuscript concerning published literature regarding model analysis of DADs. All comments of the Reviewers should be addressed in a revised version of the manuscript.

Reviewers' comments:

Reviewer's Responses to Questions

**Comments to the Author**

1. Is the manuscript technically sound, and do the data support the conclusions?

Reviewer #1: Yes

Reviewer #2: Yes

Reviewer #3: Partly

2. Has the statistical analysis been performed appropriately and rigorously?

Reviewer #1: N/A

Reviewer #2: N/A

Reviewer #3: N/A

3. Have the authors made all data underlying the findings in their manuscript fully available?

Reviewer #1: No

Reviewer #2: Yes

Reviewer #3: Yes

4. Is the manuscript presented in an intelligible fashion and written in standard English?

Reviewer #1: Yes

Reviewer #2: Yes

Reviewer #3: Yes

5. Review Comments to the Author

Reviewer #1: The paper, "Abnormal Calcium Release and Afterdepolarizations: A Comparison of Two Mathematical Models for Human Ventricular Myocytes", investigates delayed afterdepolarizations (DADs) in cardiac myocytes, which are linked to fatal arrhythmias. The study compares two human ventricular myocyte models—the ten Tusscher-Panfilov TP06 model and the HuVEC15 model. The authors analyze the effects of various electrophysiological parameters on DADs using mathematical modeling. Through continuation analysis, they reveal how calcium concentration changes in the sarcoplasmic reticulum (SR) and calcium pumps affect DADs. The study identifies three types of DADs—subthreshold, suprathreshold, and multi-blip—and uses sensitivity analysis to determine which parameters significantly influence these DAD types. Simulations demonstrate how patches of DAD cells in tissue can trigger premature ventricular complexes (PVCs), a key marker for arrhythmias. The paper emphasizes the role of the Na+/Ca2+ exchanger in suppressing DADs in the TP06 model, offering insights into potential therapeutic targets for preventing arrhythmia.

Major concerns:

1) It was not clear to me what new question this study addressed. There is an extensive body of work on model analysis of DAD’s what new question is addressed here. Please better motivate why two models were analysed and why you chose these two models.

2) Were differences explainable by underlying data or model structure?

3) There did not appear to be any validation of the models. I do not believe the TT model (not sure about HuVEC15) were designed to study DAD’s can you confirm through comparison to experimental data that they represent the key mechanism?

Query:

1. Does the Hinch model include calcium induced inactivation of the L-type Ca channel. Does this impact the generation of DADs.

2. The mesh is anisotropic with 250um edge length in plane and 1000um edge length in the z-direction. This seems very large. If the waves are flat, it may not have an impact but may be good to check.

3. The TT models are known to have a high APD depednnce on Iks, as opposed to Ikr, did this mean you needed much larger changes in TT Ikr to maintaint APD duration than HuVEC15. Could this impact your results?

4. The tissue models have each vertex representing large numbers of cells. Is this a reasonable model for DADs which are expected to be random, but in the current model you are assuming a large number of cells are synchronised, what is the mechanisms for this??

Minor

"electrogenic exchange of cytosolic Ca2+ against the extracellular Na+" → Change "against the" to "for three".

"The NCX and SERCA work together for the regulation of cytosolic Ca2+" → Consider revising "for the regulation" to "to regulate".

""calcium-overload conditions and shown the following" → Add "have" before "shown".

Reviewer #2: Here, Drs. Roshan and Pandit compare the determinants of DAD due to spontaneous calcium release events from the sarcoplasmic reticulum (SR) in two ventricular cardiomyocyte models (Ten Tusscher 2006 and Himeno HuVEC models). The authors show that SERCA and RyR play key roles in controlling SR calcium releases and that NCX plays an important role in generating DADs, with dual effects in the Ten Tusscher model. Finally, the authors demonstrate that sufficiently large areas with DADs can induce ectopic/triggered activity in multicellular simulations.

The work is largely confirmatory, corroborating previous experimental and computational studies, but is methodologically sound, mostly clearly presented, and provides an interesting overview due to its parameter sensitivity analysis.

The following aspects should be considered:

1. The initial analyses on the reduced calcium subsystem do not add much to the present manuscript. As indicated, the conditions required to generate SCRs are highly unphysiological, the method did not work in the HuVEC model and the importance of SERCA and RyR is obtained under much more relevant conditions during the subsequent analyses. Thus, the impact of this first section of the manuscript for the remainder of the story is negligible. The authors should consider removing the Section 2.1 and the corresponding methods section from the manuscript to improve focus and readability.

2. The additional spontaneous calcium releases during the AP plateau observed in the TP06 model appear rather unphysiological to this reviewer. Is there any evidence that such releases occur at these depolarized potentials? While late phase-3 EADs have obviously been documented, these tend to occur at much more negative potentials.

2b. The antiarrhythmic effects of an increase in NCX also appear to be independent of these LCRs. Indeed, since NCX primarily operates in forward mode during a normal heart beat, an increase in NCX would promote additional calcium extrusion. Thus, the sensitivity of RyR to (SR) calcium load (determined by overall calcium loading) and trigger dyadic calcium is likely a more important determinant of the effects of NCX upregulation. Indeed, previous work has shown a similar protective effect of NCX upregulation in a mouse ventricular cardiomyocyte model, independent of LCRs (Lyon et al. Front Physiol 2021).

3. Given that many studies have been conducted in this area and many of the aspects mentioned in the manuscript have been discussed previously, a better comparison to previous literature is needed. Example include, but are not limited to:

- The role of subcellular calcium releases (sparks / waves) and stochasticity in DAD generation and the limitations of deterministic common pool models are briefly mentioned in the limitations in generic terms, but key results from previous detailed models (e.g., work by Dr. Colman in PLoS Comput Biol, and studies by Dr. Sato) confirming the current results should be mentioned.

- Line 355: Indeed, the most common perception is that increased NCX would promote DADs. However, the dual role on DAD amplitude and calcium load has previously been discussed in several studies (e.g., Heijman et al. J Cardiovasc Pharmacol 2015)

- The importance of source and sink interactions could be discussed in more detail. A key manuscript in this regard is already briefly cited (ref 53), but not discussed in any detail.

Minor:

4. Please ensure that figures are presented and cited consecutively. At present the order at the end of the manuscript appears largely random. In the text, Figure 18 is cited after Figure 12 (line 368).

5. For figures 8 and 9, it would be helpful to have both in the same orientation. At present, one is in landscape and the other in portrait orientation, despite their identical structure.

6. Line 257: “we infer that the TP06 model should be capable of triggering DADs and that the addition of the RyR significantly reduces” Change to “the addition of additional RyR-mediated calcium leak significantly reduces…”

Reviewer #3: Roshan and Pandit present a simulation-based investigation on DADs in human ventricular myocytes. Many simulations results are presented, by using two computational models, and some key factors in the origin of these proarrhythmic events are highlighted.

In my view the scope of the study is not completely well defined. From the title it seems the focus would be on the comparison of two models. If so, when presenting and discussing the results it should be highlighted mainly strengths and weaknesses of the two models, which results proved to be consistent and can provide robust hypothesis about what happens in real cardiomyocyte and in case of non-concordant results between the models clarify which of the two is consistent with experiments or if this leaves open questions on the interpretation…

If, on the other hand, the aim is to add knowledge about DADs mechanisms under clcium overload and/or prevention, the open research questions could be better explicited, maybe starting from the title.

Here are a list of specific comments/observations.

Introduction:

• Line 9: human-cardiac-myocyte models: I would not talk of ‘models’ since, to my understanding, the reference is about experimental data and, in this paper, ‘model’ is always used to indicate computational/mathematical models

• Line 32-34: use of acronymous TP06 and HuVEC15 without having introduced them (only in the abstract and after at lines 45-46)

• Line 56: the choice of the models to be compared deserves some discussion. Is it possible to explain why only a few models are capable to reproduce DADs? How were the two selected among them? More recent, and likely more reliable, models of human ventricular AP have been published (e.g.: O’Hara-Rudy, Tomek-Rodriguez, Bartolucci-Passini-Severi). In particular, Bartolucci et al. claimed that their model ‘…allows reproducing delayed after-depolarizations (DADs), which could not be reproduced with the original ORd formulation’. It would have been interesting to compare this model too.

• In my opinion the overview of principal results at the end of the introduction is not needed. It would be better to further detail/clarify the objectives.

• Line 77: definition of sections 2 and 3 is quite superimposed, maybe here ‘presentation’ instead of ‘discussion’?

Methods

• A very short description of Markov model used in TP06 to model RyR would be useful

• Line 97: definition of CaRU is missing

• Line 104: please make it explicit that the model is an isotropic monodomain.

• Line 114-16: why different solvers for Vm and gating variables?

• Line 119: HUVEC15 capital U here

• Line 125: 2cm of thickness seems much larger than any cardiac tissue

• Why in figure 2 is the R4 case not reported? I suggest to add in the figure legend R1-R4 to better identify the 4 cases

• Figure 2: the actual increase (overload) of calcium seems quite small in the HuVEC15 model. Does this mean that in this model DADs are not due to calcium overload?

• Line 165: which Appendix? Why only TP06?

• Section 1.4: What is the value of the analysis of a calcium subsystem in which the main calcium current (ICaL) is neglected and INaCa artificially reversed by setting completely unphysiological sodium concentrations? It is not clear to me which results of this analysis have been used to inform the subsequent ones. If anything is really strictly required, I would withdraw this part.

Results

• Figure legends are missing! (at least in the pdf provided to reviewers)

• Line 219: Ca2+ subsystem (see Subsec. 2.1): but we are in Subsec 2.1 should it be Subsec 1.4?

• Section 2.1: see comment above. How can this analysis provide strong support for the occurrence DADs? DADs are, by definition, AFTERdepolarizations, occurring ‘after’ normal depolarizations. An autonomous system, with no external stimulation and no normal depolarizations is a completely different system.

• Section 2.2: this is a bit confusing. How were selected the parameters to be tuned? Which values were used to obtain different DADs types? Apparently this should go after the results of sensitivity analysis.

• Section 2.3: It seems to me that the most important ‘output’ with respect to which the sensitivity analysis should be performed is the occurrence of DADs, rather than their frequency and amplitude (although amplitude is relevant too). A logistic regression approach would be more appropriate then (see other publications by the Sobie’s Lab)

• Line 292: in HuVCE15 Vrel is the most sensitive parameter for DAD frequency, but from fig.6 it seems to be GCaL.

• Line 310-316: I cannot follow the description of this result, find where the ‘biphasic’ dependence on SKNaCa is shown, etc. Please double check!

• In figure 8(b) it is not clear what is reported into the x-axis, since above it is reported SKNaCa=2

• Line 405, description of figure 14 seems to not correspond to the figure itself

• Section 2.7: all the 1D/2D/3D simulations are carried out (if I understood correctly) assuming groups of identical cells, close each other, generating exactly the same kind of DADs at exactly the same time, when simulated as isolated cells. This obviously makes synchronization and propagation of DADs much more likely than in real tissues where a huge heterogeneity is in place.

Discussion

• Line 446: capital U

• Line 477: delete for

• Line 515: are figures shown for only the case of clumps of one DAD type?

• Line 532: what is Paper II?

• Line 540-546: please split the sentence to make it more readable

APPENDIX

• Eq 5. In Appendix: is it different from the original TP06 formulation?

• Line 711: delete Ref and “U” in HUVEC15

• Description of video 3: Line 590-91 – is there an extra sentence?

• Line 700: Vrel*0.000075ms^-1???

6. PLOS authors have the option to publish the peer review history of their article (what does this mean?). If published, this will include your full peer review and any attached files.

Reviewer #1: No

Reviewer #2: No

Reviewer #3: No

---

## [Author Response · Author response to Decision Letter 1]

28 Dec 2024

Dear Editors,

Thank you for your message and for the reports of the Referees on our paper. We thank them for taking the time to prepare their reports. We provide, below this message, (a) our answers to all the questions and comments in the reports of the Referees and (b) a list of all the changes in our revised manuscript (we have made these changes to address the points raised by the Referees). Our detailed responses to all the queries and comments of the Referees are given below.

As you can see from our responses, all the points raised by the Referees have been addressed by us.

We hope therefore that our paper will now be accepted for publication in Plos One.

Please note that “The funders had no role in study design, data collection and analysis, decision to publish, or preparation of the manuscript”.

The information regarding the funding resources we have moved it from the manuscript to the online submission form.

These are the funding agencies:

1. NSM: DST/NSM/R&D_HPC_Applications/Extension/2023/09

2. National Science Chair: National Science Chair NSC/2022/000020

Thanking you and with our best regards,

Yours sincerely,

Navneet Roshan

and Rahul Pandit

---

## [Decision Letter · Decision Letter 1]

23 Mar 2025

PONE-D-24-32626R1Abnormal Calcium Release and Afterdepolarizations: A Comparison of Two Mathematical Models for Human Ventricular MyocytesPLOS ONE

Dear Dr. Roshan,

Thank you for submitting your manuscript to PLOS ONE. After careful consideration, we feel that it has merit but does not fully meet PLOS ONE’s publication criteria as it currently stands. Therefore, we invite you to submit a revised version of the manuscript that addresses the points raised during the review process.

Although the manuscript has been improved upon since its previous iteration, there are still a number of issues that need to be discussed, as highlighted by the comments of Reviewer 1. For example, an increase in the clarity of the hypothesis as well as a more methodological approach to certain parts of the manuscript. All comments would need to be addressed in a revised version of the manuscript

We look forward to receiving your revised manuscript.

Kind regards,

Daniel M. Johnson, PhD

Academic Editor

PLOS ONE

Reviewers' comments:

Reviewer's Responses to Questions

**Comments to the Author**

1. If the authors have adequately addressed your comments raised in a previous round of review and you feel that this manuscript is now acceptable for publication, you may indicate that here to bypass the “Comments to the Author” section, enter your conflict of interest statement in the “Confidential to Editor” section, and submit your "Accept" recommendation.

Reviewer #1: (No Response)

Reviewer #2: All comments have been addressed

2. Is the manuscript technically sound, and do the data support the conclusions?

Reviewer #1: Yes

Reviewer #2: Yes

3. Has the statistical analysis been performed appropriately and rigorously?

Reviewer #1: N/A

Reviewer #2: N/A

4. Have the authors made all data underlying the findings in their manuscript fully available?

Reviewer #1: Yes

Reviewer #2: Yes

5. Is the manuscript presented in an intelligible fashion and written in standard English?

Reviewer #1: Yes

Reviewer #2: Yes

6. Review Comments to the Author

Reviewer #1: Abnormal Calcium Release and Afterdepolarizations: A Comparison of Two Mathematical Models for Human Ventricular Myocytes

This study uses computational simulations to investigate the electrophysiological conditions leading to delayed afterdepolarisations (DADs) in the human left ventricle. This effect is demonstrated using two different ODE-based models derived from published models by Ten Tusscher (TP06) and Himeno (HuVEC15). The authors detect changes in the DAD activity predicted by these models (represented using “phase diagrams”) as model parameters are altered. They then interpret these observations in terms of the mechanisms underlying DAD formation, which are associated with intracellular calcium dynamics. The implications of these behaviours is explored in multi-dimensional finite-element simulations.

OVERALL OBSERVATION

The subject matter of the paper is certainly interesting and I agree that a mechanistic understanding of DADs is useful, e.g., for the development of pharmacological treatments. The topic lends itself well to computational simulation, to resolve the roles played by the different system components and to explore hypothetical scenarios, as was done in this study. However, I feel that the study, as presented here, has significant shortcomings that prevent its conclusions from being truly convincing. I therefore cannot recommend this manuscript for publication without major improvements.

In summary, my objections are along the following general themes:

1. lack of a clearly formulated hypothesis to test. It is established from the outset that DADs arise from calcium dynamics. The aim of the paper is hence defined as “to delineate the role of the parameters on the DAD features [using two ventricular models]”, but I find this somewhat vague. The paper thus reads like “we’ll test several models and see if anything interesting comes out”. This lack of clarity of purpose transpires in the Discussion, where conclusions are drawn with regard to one model (the amended TP06) but not the other. Because the outcomes of the different models differ significantly, it is difficult to draw a convincing overall conclusion and to make a critical comparison of the models in the perspective of the real system. A clearer, more focused mission statement would help justify the choice of model(s) and tailor the analysis methodology more systematically.

2. questionable methodology and analysis. The main conclusions of the study rest on the interpretation of the “phase diagrams”, which arguably segregate the parameter spaces into qualitatively distinct regimes. As explained specifically below, I have reservations with regards to both (i) the definitions of the different phases and (ii) the analysis and conclusions drawn from the (arbitrarily chosen?) set of sections through the phase space. Although I appreciate the difficulty of rigorously defining regimes of behaviour, these challenges must be addressed critically for the results to be instructive.

3. overambitious scope. Although understanding DADs at the cell level must ultimately benefit understanding at the tissue and organ levels, combining all these levels within this study seems to me to be premature, especially as so many questions remain to be addressed at the basic level. To avoid confusion, I would encourage the authors to proceed more methodically before attempting to tackle the multi-scale problem.

SPECIFIC CRITIQUES

Model choice and constitution

• The exact rationale for choosing TP06 and HuVEC15, as opposed to the other candidates, is not clearly and explicitly explained. The “Ca2+-oscillation hypotheses” are cited but without an explanation of the underlying principle. Despite similarities, the two chosen models differ markedly in terms of their structure and predictions. The reader may then wonder whether several models were used in the hope that one would yield appealing conclusions.

• The ability of TP06 to produce DADs relies on an alteration of the published model but the nature of that change is not explained or supported. The reader is directed to the Supplement, which I don’t feel is appropriate, given the centrality of this issue to the paper. Without a proper clarification, justification, and validation of the alterations, it is difficult to establish that TP06 is a suitable platform for the present purpose. However, we learn only in the Discussion section that the TP06 amendment involves introducing a SR leak current. It is potentially misleading to even refer to the altered model as “TP06”. Furthermore, as shown in Fig. 3, the DADs in the TP06 case coincide with the appearance of additional oscillations within the AP that, in the absence of explicit commentary, raises questions on the reliability of the model. Was this model included in the study simply because its apparent demonstration of the “protective” effect of NCX is appealing?

• In the case of HuVEC15, the NCX distribution was adjusted, “to increase the amplitude of the DADs” (line 162). What were the targets? Is this recalibration of the original model justified? Can we be certain that the “phase diagrams” do not depend critically on the altered model calibrations?

Analysis and interpretations.

• The two models TP06 and HuVEC15 are described in parallel rather than in sequence (Sec 2.1). Details of numerical integration are also given in parallel (Sec 2.2). I think it would help the logical flow if they weren’t so merged. No explanation is given as to why the analysis protocols differ between the two models. For example, TP06 is solved using the Rush-Larsen schem and HUVEC15 using the generalized Rush-Larsen scheme (line 108). What motivated these choices? Can we be certain that the differences in outcomes arise from intrinsic biophysical differences between the models rather than from details in their implementation?

• Ca2+ overload is implemented by varying GCaL while also varying GKr to keep the APD constant. Was the impact of other model parameters on the APD (and indeed on the AP morphology) considered when choosing GKr as the correction factor? The AP morphology in Fig. 2 changes noticeably with GCaL, despite the GKr correction. Might the subsequent conclusions be affected by the precise definition of APD adopted?

• Given the centrality of the “frequency” and “amplitude” values in the generation of the phase diagrams, an appraisal of these quantites in terms of real-life DADs would be useful, if only to help justify the chosen model alterations.

More specifically, Fig. 3 shows that both the time period and amplitudes of consecutive DADs are generally not constant. How does this compare to the real-life case? Are the amplitude and frequency values, “averaged” over (presumably?) the steady-state even meaningful characterisations?

• I do not think that defining the DAD frequency as equal to zero in the supra-threshold regime is meaningful; it can even be misleading. Likewise, what is the meaning of “frequency” in the case where only one (or even two!) DAD occurs? The concept of frequency really only makes sense in the context of regular cycling, but it loses clarity when the number of full cycles is very small (at most two in the present case). The “multi-blip” DADs are presented as a class distinct from the single-blip DADs. Is this qualitative distinction objectively justified, or could they simply result from the time difference between consecutive blips simply exceeding the pacing period? These uncertainties can generate scepticism on the interpretations of the “phase transitions” in Figs 6 and 7.

Is the casting of these transitions as “Hopf bifurcations”, mentioned in the Discussion section for the first time, properly supported?

• Sensitivity analysis results Sec 3.2. Line 240 reads “for the HuVEC15 model, Vrel is the most sensitive parameter for the DAD frequency”. This doesn’t seem right, as the GCaL sensitivity is approximately twice as large, and KNaCa is negative and of similar magnitude.

• The conclusion of results Sec 3.3 is the “important difference” between the TP06 and HuVEC15 results with regard to the reentry of the “normal AP” phase at large SKNaCa. However, I do not find this conclusion very convincing. Although Figs 6 and 7 are described as “representative sections” of the parameter space, there is no indication that they provide an objective overall understanding of the entire space. How “representative” are these sections?

It is stated that there is such a reentry for TP06 but not HuVEC15, but such reentry is not apparent in Fig. 6d. Can we have confidence in these statements, based on the very coarsely pixelated phase diagrams of Figs. 6 and 7? If the occurrence of a transition is apparently so dependent on the trajectory in the parameter space, can we be certain that the TP06 and HuVEC15 differ fundamentally (line 314: “this protective mechanism is only present in the TP06 model”)?

Reviewer #2: I thank the authors for addressing my comments in this revised version. I do not have any additional suggestions.

7. PLOS authors have the option to publish the peer review history of their article (what does this mean?). If published, this will include your full peer review and any attached files.

Reviewer #1: No

Reviewer #2: No

---

## [Decision Letter · Decision Letter 2]

1 Dec 2025

Abnormal Calcium Release and Afterdepolarizations: A Comparison of Two Mathematical Models for Human Ventricular Myocytes

PONE-D-24-32626R2

Dear Dr. Roshan,

We’re pleased to inform you that your manuscript has been judged scientifically suitable for publication and will be formally accepted for publication once it meets all outstanding technical requirements.

Kind regards,

Elena G. Tolkacheva, PhD

Academic Editor

PLOS ONE

Additional Editor Comments (optional):

Reviewers' comments:

Reviewer's Responses to Questions

**Comments to the Author**

1. If the authors have adequately addressed your comments raised in a previous round of review and you feel that this manuscript is now acceptable for publication, you may indicate that here to bypass the “Comments to the Author” section, enter your conflict of interest statement in the “Confidential to Editor” section, and submit your "Accept" recommendation.

Reviewer #2: All comments have been addressed

2. Is the manuscript technically sound, and do the data support the conclusions?

Reviewer #2: (No Response)

3. Has the statistical analysis been performed appropriately and rigorously?

Reviewer #2: (No Response)

4. Have the authors made all data underlying the findings in their manuscript fully available?

Reviewer #2: (No Response)

5. Is the manuscript presented in an intelligible fashion and written in standard English?

Reviewer #2: (No Response)

6. Review Comments to the Author

Reviewer #2: (No Response)

7. PLOS authors have the option to publish the peer review history of their article (what does this mean?). If published, this will include your full peer review and any attached files.

Reviewer #2: No

---

## [Editor Report · Acceptance letter]

PONE-D-24-32626R2

PLOS One

Dear Dr. Roshan,

I'm pleased to inform you that your manuscript has been deemed suitable for publication in PLOS One. Congratulations! Your manuscript is now being handed over to our production team.

Kind regards,

on behalf of

Dr. Elena G. Tolkacheva

Academic Editor

PLOS One